# Counterfactual Data Augmentation with Contrastive Learning

## Abstract

Statistical disparity between distinct treatment groups is one of the most significant challenges for estimating Conditional Average Treatment Effects (CATE). To address this, we introduce a model-agnostic data augmentation method that imputes the counterfactual outcomes for a selected subset of individuals. Specifically, we utilize contrastive learning to learn a representation space and a similarity measure such that in the learned representation space *close* individuals identified by the learned similarity measure have *similar* potential outcomes. This property ensures reliable imputation of counterfactual outcomes for the individuals with close neighbors from the alternative treatment group. By augmenting the original dataset with these reliable imputations, we can effectively reduce the discrepancy between different treatment groups, while inducing minimal imputation error. The augmented dataset is subsequently employed to train CATE estimation models. Theoretical analysis and experimental studies on synthetic and semi-synthetic benchmarks demonstrate that our method achieves significant improvements in both performance and robustness to overfitting across state-of-the-art models.

## 1 Introduction

One of the most significant challenges for Conditional Average Treatment Effect (CATE) estimation is the statistical disparity between distinct treatment groups (Goldsmith-Pinkham et al., 2022). While Randomized Controlled Trials (RCT) mitigate this issue (Rubin, 1974; Imbens & Rubin, 2015), they can be expensive, unethical, and sometimes unfeasible to conduct. Consequently, we are often constrained to relying on observational studies, which are susceptible to selection bias arising from the aforementioned issue. To address this, we introduce a model-agnostic data augmentation method, comprising two key steps. First, our approach identifies a subset of individuals whose counterfactual outcomes can be reliably imputed. Subsequently, it performs imputation for the counterfactual outcomes of these selected individuals, thereby augmenting the original dataset with the imputed values. Importantly, our method serves as a data pre-processing module that remains *agnostic to the choice of the subsequent model* employed for CATE estimation. Extensive experiments underscore the efficacy of our approach, as it consistently delivers substantial performance improvements across various models, including state-of-the-art models for CATE estimation. Furthermore, our method has been empirically validated to effectively mitigate overfitting, a significant challenge in CATE estimation applications due to the inherent inaccessibility of counterfactual data.

Our method is motivated by an observed trade-off between *(i)* the discrepancy across treatment groups and *(ii)* the error induced by the imputation of the missing counterfactual outcomes. Consider the scenario with one control group and one treatment group. In this context, no individual can appear in both the control and treatment groups due to the inaccessibility of counterfactual outcomes (Holland, 1986). To illustrate the core concept behind our methodology, consider the following experiment: for individuals in the control group (and reciprocally, the treatment group), we randomly impute their outcome under treatment (or reciprocally, in the absence of treatment), generating their counterfactual outcomes. Subsequently, we integrate each individual along with their imputed outcomes back into the dataset. This transformation ensures that individuals from both the control and treatment groups become identical, effectively eliminating any disparities. However, it becomes evident that any model trained on such a randomly augmented dataset would exhibit poor performance, primarily due to the substantial error introduced by the random imputation of the counterfactual outcomes. Here, our approach seeks to address this challenge by identifying a *subset* of individuals for

whom the counterfactual outcomes can be reliably imputed. Hence, *the positive impact of disparity reduction will outweigh the negative impact of imputation error*. While it's important to acknowledge that the inherent issue of CATE estimation cannot be entirely eliminated, our approach effectively alleviates the challenges faced by CATE estimation models, thus facilitating their learning process and improving their performance.

In this paper, we utilize contrastive learning to identify the individuals whose counterfactual outcomes can be reliably imputed. This technique helps us develop a representation space and a similarity measure, such that within this learned representation space, *close* individuals by the similarity measure exhibit *similar* potential outcomes. This *smoothness* property guarantees highly reliable imputation of counterfactual outcomes for the individuals with a sufficient number of close neighbors from the *alternative* treatment group. Specifically, we impute the counterfactual outcomes for these individuals by utilizing the factual outcomes of their proximate neighbors. Importantly, this smoothness property ensures that the imputation can be achieved locally with simple models that require minimal tuning. We explore two distinct methods for imputation: linear regression and Gaussian Processes.

To comprehensively assess the efficacy of our data augmentation technique, we demonstrate that our approach asymptotically generates datasets whose probability densities converge to those of RCTs. In addition, we provide non-asymptotic generalization bounds for the performance of CATE estimation models trained with our augmented data. Our empirical results further demonstrate the efficacy of our method, showcasing consistent enhancements in the performance of state-of-the-art CATE estimation models, including TARNet, CFR-Wass, and CFR-MMD (Shalit et al., 2017), S-Learner and T-Learner integrated with neural networks, Bayesian Additive Regression Trees (BART) (Hill, 2011; Chipman et al., 2010; Hill et al., 2020) with X-Learner (Künzel et al., 2019), and Causal Forests (CF) (Athey & Imbens, 2016) with X-Learner.

**Related Work.** One of the fundamental tasks in causal inference is to estimate *Average Treatment Effects* (ATE) and *Conditional Average Treatment Effects* (CATE) (Neyman, 1923; Rubin, 2005). Various methods have been proposed to address ATE estimation task, including Covariate Adjustment (Rubin, 1978), Propensity Scores (Rosenbaum & Rubin, 1983), Doubly Robust estimators (Funk et al., 2011), Inverse Probability Weighting (Hirano et al., 2003), and recently Reisznet (Quintas-Martinez et al., 2021). While these methods are successful for ATE estimation, they are not directly applicable to CATE estimation. Recent advances in machine learning have led to new approaches for CATE estimation, such as decision trees (Athey & Imbens, 2016), Gaussian Processes (Alaa & Van Der Schaar, 2017), Multi-task deep learning ensemble (Jiang et al., 2023), Generative Modeling (Yoon et al., 2018), and representation learning with deep neural networks (Shalit et al., 2017; Johansson et al., 2016). It is worth noting that alternative approaches for investigating causal relationships exist, such as graphical modeling and do-calculus, as proposed by Pearl (Pearl, 2009a;b). In this work, we adopt the Neyman-Rubin framework. At its core, the CATE estimation problem can be seen as a missing data problem (Rubin, 1974; Holland, 1986; Ding & Li, 2018) due to the unavailability of the counterfactual outcomes. In this context, we propose a new data augmentation approach for CATE estimation by imputing certain missing counterfactuals. Data augmentation, a well-established technique in machine learning, serves to enhance model performance and curb overfitting by artificially expanding the size of the training dataset (Van Dyk & Meng, 2001; Chawla et al., 2002; Han et al., 2005; Jiang et al., 2020; Chen et al., 2020a; Liu et al., 2020; Feng et al., 2021). A crucial aspect of our methodology is the identification of similar individuals. There are various methods to achieve this goal, including propensity score matching (Rosenbaum & Rubin, 1983), and Mahalanobis distance matching (Imai et al., 2008). Nonetheless, these methods pose significant challenges, particularly in scenarios with large sample sizes or high-dimensional data, where they suffer from the curse of dimensionality. Recently, Perfect Match (Schwab et al., 2018) is proposed to leverage importance sampling to generate replicas of individuals. It relies on propensity scores and other feature space metrics to balance the distribution between the treatment and control groups during the training process. In contrast, we utilize contrastive learning to construct a similarity metric within a representation space. Our method focuses on imputing missing counterfactual outcomes for a selected subset of individuals, without creating duplicates of the original data points. While the Perfect Match method is a universal CATE estimator, our method is a model-agnostic data augmentation method that serves as a data preprocessing step for other CATE estimation models. It is important to note that in recent years, other works on data augmentation for the intersection of domain generalization and causal inference have been proposed (Ilse et al., 2021; Mahajan et al., 2021).

## 2 THEORETICAL BACKGROUND

Let $T \in \{0, 1\}$ be a binary variable for treatment assignment, $X \in \mathcal{X} \subset \mathbb{R}^d$ be the covariates (features), and $Y \in \mathcal{Y} \subset \mathbb{R}$ be the factual (observed) outcome. For each $j \in \{0, 1\}$, we define $Y_j$ as the *potential outcome* (Rubin, 1974), which represents the outcome that would have been observed if only the treatment $T = j$ was administered. The random tuple $(X, T, Y)$ jointly follows the *factual (observational) distribution* denoted by $p_F(x, t, y)$. The dataset used for causal inference, denoted by $D_F = \{(x_i, t_i, y_i)\}_{i=1}^n$, consists of $n$ observations independently sampled from $p_F$ where $n$ is the total number of observations. The counterfactual distribution, denoted by $p_{CF}$, is defined as the sampling distribution of the dataset in a hypothetical parallel universe where the treatment assignment mechanism is inverted. To simplify the notation, for any distribution $p(x, t, y)$, we use $p(x, t)$ (respectively, $p(x)$) to denote the marginalized distribution of $p(x, t, y)$ over the random tuple $(X, T)$ (respectively, $X$). For example, $p_F(x, t)$ is the factual joint distribution of $X$ and $T$. For a binary treatment assignment, the following identity holds: $p_{CF}(x, 1 - t) = p_F(x, t)$ (Shalit et al., 2017; Peters et al., 2017).

**Definition 1** (CATE). *The Conditional Average Treatment Effect (CATE) is defined as the expected difference in potential outcomes given the covariates $X = x$:*

$$\tau(x) = \mathbb{E}[Y_1 - Y_0 | X = x]. \tag{1}$$

**Definition 2** (ATE). *The Average Treatment Effect (ATE) is defined as:*

$$\tau_{ATE} = \mathbb{E}[Y_1 - Y_0]. \tag{2}$$

CATE and ATE are identifiable under the assumptions of *positivity*, i.e., $0 < p_F(T = 1|X) < 1$, and *conditional unconfoundedness*, i.e., $(Y_1, Y_0) \perp\!\!\!\perp T|X$ (Robins, 1986; Imbens & Rubin, 2015). Let $\hat{\tau}(x) = h(x, 1) - h(x, 0)$ denote an estimator for CATE where $h$ is a hypothesis $h : \mathcal{X} \times \{0, 1\} \to \mathcal{Y}$ that estimates the underlying causal relationship between $(X, T)$ and $Y$.

**Definition 3.** *The Expected Precision in Estimating Heterogeneous Treatment Effect (PEHE) (Hill, 2011) is defined as:*

$$\varepsilon_{PEHE}(h) = \int_{\mathcal{X}} (\hat{\tau}(x) - \tau(x))^2 p_F(x)dx = \int_{\mathcal{X}} (h(x, 1) - h(x, 0) - \tau(x))^2 p_F(x)dx. \tag{3}$$

**Definition 4.** *Given a joint distribution $p$ over $(X, T, Y)$ and a hypothsis $h : \mathcal{X} \times \{0, 1\} \to \mathcal{Y}$, let $\mathcal{L}_p(h)$ be defined as:*

$$\mathcal{L}_p(h) = \int (y - h(x, t))^2 p(x, t, y) \, dx \, dt \, dy.$$

*Then the factual loss $\mathcal{L}_F$ and the counterfactual loss $\mathcal{L}_{CF}$ are respectively defined as:*

$$\mathcal{L}_F(h) = \mathcal{L}_{p_F}(h), \quad \mathcal{L}_{CF}(h) = \mathcal{L}_{p_{CF}}(h) \tag{4}$$

**Remark.** $\varepsilon_{PEHE}$ is widely-used as the performance metric for CATE estimation. However, estimating $\varepsilon_{PEHE}$ directly from observational data such as $D_F$ is a non-trivial task, as it requires knowledge of the counterfactual outcomes to compute the ground truth CATE values. This inherent challenge underscores that models for CATE estimation need to be robust to overfitting the factual distribution. Our empirical results (see Section 5) indicate that our method mitigates the risk of overfitting for various CATE estimation models.

## 3 COCOA: CONTRASTIVE COUNTERFACTUAL AUGMENTATION

As discussed in Section 1, the effectiveness of imputing counterfactual outcomes depends on the availability of a representation space and a similarity measure that satisfies a criterion: within this representation space, individuals identified as similar by the similarity measure should exhibit similar potential outcomes when subjected to the same treatment. This smoothness assumption ensures straightforward local approximation: *an individual's potential outcomes should exhibit a strong correlation with those of its nearby neighbors*. In essence, for the individuals who possess a sufficient number of close neighbors from the alternative treatment group, we can impute their *counterfactual outcomes* using the *factual outcomes* of their nearby neighbors.

To this end, we propose _COntrastive COunterfactual Augmentation_ (COCOA) consisting of two key components. The first component is a classifier $g_{\theta^*}$, trained using contrastive learning (Le-Khac et al., 2020; Jaiswal et al., 2020) to predict whether two individuals have similar outcomes when subjected to the same treatment. The second component is a local regressor $\psi$, which imputes the counterfactual outcome for a given individual after being fitted to the close neighbors. The pseudo-code of COCOA is illustrated in Algorithm 1. Particularly, the trained classifier $g_{\theta^*}$ first identifies the close neighbors of a given individual. These close neighbors are individuals who are likely to have similar potential outcomes. Subsequently, the non-parametric regressor $\psi$ utilizes the factual outcomes of these identified individuals to estimate the counterfactual outcome of the target individual. For example, if the target individual $x$ belongs to the control group, we employ $g_{\theta^*}$ to select a subset of individuals from the treatment group who may exhibit outcomes similar to those of $x$. Subsequently, $\psi$ uses the factual outcomes of these selected individuals from the treatment group to estimate the counterfactual outcome of $x$ under treatment. Finally, the imputed outcome of $x$ is incorporated into the original dataset. As discussed in Section 1, the minimal error of the counterfactual imputation plays a crucial role in the success of our method. To ensure the reliability of these imputations, we only perform imputations for individuals who possess a sufficient number of close neighbors. In our experiments, we set the minimum required number of close neighbors to be 5. In the worst case, no individuals will meet these criteria for imputation, resulting in no augmentation of the dataset. This approach guarantees that our method does not compromise the information inherent in the original dataset.

**Remark.** COCOA differs from standard CATE estimation models: it _does not generalize_ to unseen samples. Instead, its objective is to identify individuals within a given dataset and then impute their counterfactual outcomes. In essence, COCOA serves as a tool to augment the dataset for other CATE models, thereby improving their ability to make accurate predictions on unseen samples.

## 3.1 CONTRASTIVE LEARNING MODULE

Contrastive (representation) learning methods (Wu et al., 2018; Bojanowski & Joulin, 2017; Dosovitskiy et al., 2014; Caron et al., 2020; He et al., 2020; Chen et al., 2020b; Trinh et al., 2019; Misra & Maaten, 2020; Tian et al., 2020) are based on the fundamental principle that similar individuals should be associated with closely related representations within an embedding space. This is achieved by training models to perform an auxiliary task: predicting whether two individuals are similar or dissimilar. The definition of similarity often depends on the context of the downstream task. In the context of CATE estimation, we consider two individuals with _similar outcomes_ under the same treatment as _similar individuals_. The degree of similarity between outcomes is measured using a particular metric in the potential outcome space $\mathcal{Y}$. In our case, we employ the Euclidean norm in $\mathbb{R}^1$ for this purpose. With this perspective, given the factual (original) dataset $D_{\mathrm{F}} = \{(x_i, t_i, y_i)\}_{i=1}^n$, we construct a _positive dataset_ $D^+$ that includes pairs of similar individuals. Specifically, we define $D^+ = \{(x_i, x_j) : i, j \in [n], i \neq j, t_i = t_j, \|y_i - y_j\| \leq \epsilon\}$ where $\epsilon$ is user-defined sensitivity parameter specifying the desired level of precision. We also create a _negative dataset_ $D^- = \{(x_i, x_j) : i, j \in [n], i \neq j, t_i = t_j, \|y_i - y_j\| > \epsilon\}$ containing pairs of individuals deemed dissimilar. Let $\ell : \{0, 1\} \times \{0, 1\} \to \mathbb{R}$ be any loss function for classification task. We train a parametric classifier $g_\theta : \mathcal{X} \times \mathcal{X} \to \mathbb{R}$ where $\theta$ is the parameter vector that minimizes:

$$\theta^* \in \arg\min_\theta \sum_{(x,x') \in D^+} \ell(g_\theta(x, x'), 1) + \sum_{(x,x') \in D^-} \ell(g_\theta(x, x'), 0). \tag{5}$$

**Selection of the Nearest Neighbors for imputation.** For a given individual $x$ with treatment $t$, We utilize $g_{\theta^*}$ to identify a group of individuals from the factual dataset suitable for counterfactual imputation. Specifically, we iterate over all the individuals who received treatment $1 - t$ and employ $g_{\theta^*}$ to predict whether their potential outcomes are close to the potential outcome of $x$ under treatment $1 - t$. Let $x$ be the individual who received treatment $t$ and whose counterfactual outcome will be imputed. Its selected neighbors $D_{x,t}$[1] is defined as:

$$D_{x,t} \doteq \{i \in [n] : t_i = 1 - t, g_{\theta^*}(x, x_i) = 1\} \tag{6}$$

To ensure the quality of counterfactual imputation, we only estimate the counterfactual outcome of $x$ if $|D_{x,t}| \geq K$, where $K$ is a pre-determined parameter that controls estimation accuracy. In essence,

---

[1]The terms "individual" and "indices of individuals" are used interchangeably.

---

**Algorithm 1:** COCOA: Contrastive Counterfactual Augmentation

---

**Input:** Factual dataset $D_\mathrm{F} = \{(x_i, t_i, y_i)\}_{i=1}^n$, $n$ is the total number of samples; sensitivity
parameter $\epsilon$; threshold $K$ for selecting individuals to augment counterfactuals

**Output:** Augmented factual dataset $D_\mathrm{AF}$ as training data for CATE estimation models

1 **Function** `Main`:

2      $D^A \leftarrow \emptyset$          ▷ **Step 1: Train the contrastive learning module**

3      Construct two datasets $D^+$ and $D^-$      (See Section 3.1 for details)

4      Train a parametric model $g_\theta$, where $\theta$ is the parameter vector, by optimizing:
         $\theta^* \in \arg\min_\theta \sum_{(x,x') \in D^+} \ell(g_\theta(x, x'), 1) + \sum_{(x,x') \in D^-} \ell(g_\theta(x, x'), 0)$
         ▷ **Step 2: Augment the factual dataset**

5      **for** $i = 1, 2, \ldots, n$ **do**

6          $N_i \leftarrow \{(x_j, y_j) | j \in [n], t_j = 1 - t_i, g_{\theta^*}(x_i, x_j) = 1\}$    (Only augment the factual data if the number

                                                   of close neighbors is sufficient)

7          **if** $|N_i| \geq K$ **then**

8              $\hat{y}_i \leftarrow \psi(x_i, N_i)$ $D^A \leftarrow D^A \cup \{(x_i, 1 - t_i, \hat{y}_i)\}$

9      $D_\mathrm{AF} \leftarrow D^A \cup D_\mathrm{F}$

10      **return** $D_\mathrm{AF}$

---

our approach ensures that we augment the counterfactual outcome for an individual solely when there exists a sufficient number of closely related neighbors within the dataset.

## 3.2 LOCAL REGRESSION MODULE

After identifying the nearest neighbors, we employ a local regression module $\psi$ to impute the counterfactual outcomes. $\psi$ has two arguments *(i)* $x$: the individual whose counterfactual outcome needs to be imputed; *(ii)* $D_x = \{(x_i, y_i)\}_{i=1}^{n_x}$: a set of close neighbors to $x$ from the alternative treatment group with $y_i$ are their factual outcomes and $n_x$ is the number of close neighbors. In this work, we explore two different types of local regression modules which are linear regression and Gaussian Process (GP). In experimental studies, we present results with GP using a DotProduct Kernel and defer the results for other kernels and linear regression to Appendix C.2 due to space limitation. We opt for these relatively straightforward function classes for local regression motivated by the following three principles:

1. *Local Approximation*: complex functions can be locally estimated with simple functions, e.g., continuous functions and complex distributions can be approximated by a linear function (Rudin, 1953) and Gaussian distributions (Tjøstheim et al., 2021), respectively.

2. *Sample Efficiency*: if the class of the local linear regression module can estimate the true target function locally, then a class with less complexity will require fewer close neighbors for good approximations.

3. *Practicality*: A simpler class of $\psi$ requires less hyper-parameter tuning which is even more challenging in causal inference applications.

**Gaussian Process.** Gaussian Processes (Seeger, 2004) offers a robust non-parametric approach to regression. Consider $\phi(x) \sim \mathcal{GP}(m(x), k(x, x'))$, for $x, x' \in \mathcal{X}$. It is a collection of random variables indexed by a set $\mathcal{X}$ such that any finite collection of them follows a multivariate Gaussian distribution. Consider a finite index set of $n$ elements $X \doteq \{x_i\}_{i=1}^n$, then the $n$-dimensional random variable $\phi(X) \doteq [\phi(x_1), \phi(x_2), \ldots, \phi(x_n)]$ follows a Gaussian distribution:

$$\phi(X) \sim \mathcal{N}(m(X), K(X)) \tag{7}$$

where $m(X) \doteq [m(x_1), \ldots, m(x_n)]$ is the mean and the $K(X)$ is a $n \times n$ covariance matrix whose element on the $i$-th row and $j$-th column is defined as $K(X)_{ij} \doteq K(x_i, x_j)$. From a functional perspective, a GP imposes a prior over functions $\phi : \mathcal{X} \to \mathbb{R}$, which is completely characterized by a mean function $m : \mathcal{X} \to \mathbb{R}$ and a kernel $k : \mathcal{X} \times \mathcal{X} \to \mathbb{R}$. $m$ and $K$ encapsulate prior beliefs about the smoothness and periodicity of $\phi$.

Based on the principle of Local Approximation, we assume that, given the construction of $D_{x,t}$, the counterfactual outcome of an individual $x$ and the factual outcomes of individuals within $D_{x,t}$ follow a GP. If $x$ received treatment $t$, after locating the group of close neighbors using the method described in Section 3.1, the imputed counterfactual outcome for $x$ is described as follows:

$$\hat{y}_x^{1-t} \doteq \mathbb{E}[y^{1-t}|x, \{y_i^{1-t}\}], \quad i \in D_{x,t} \tag{8}$$

Let $\sigma(i)$ denote the $i$-th smallest index in $D_{x,t}$ and $k$ denote the kernel (covariance function) of GP. The imputed counterfactual outcome will have the following close-form formula:

$$\hat{y}_x^{1-t} = \mathbb{E}[y^{1-t}|x, y_i^{1-t}, i \in D_{x,t}] = K_x^T K_{xx}^{-1} y \tag{9}$$

where $y = [y_{\sigma(1)}^{1-t}, \ldots, y_{\sigma(|D_{x,t}|)}^{1-t}]$, $K_x = [k(x, x_{\sigma(1)}), \ldots, k(x, x_{\sigma(|D_{x,t}|)})]$ and $K_{x,x}$ is a $|D_{x,t}| \times |D_{x,t}|$ matrix whose element on the $i$-th row and $j$-column is $k(x_{\sigma(i)}, x_{\sigma(j)})$. Finally, we append the tuple $(x, 1-t, \hat{y}_x^{1-t})$ into the factual dataset to augment the training data.

# 4 THEORETICAL ANALYSIS

In this section, we present two main theoretical results to support the efficacy of our approach and to provide a deeper understanding of its robustness. Our first result characterizes the asymptotic behavior of data augmentation methods for causal inference, demonstrating that the distribution of the augmented dataset converges towards the distribution of randomized controlled trials (RCTs) - the gold standard for causal inference applications. Our second result establishes a generalization bound for the performance of CATE estimation models trained using the augmented dataset. Importantly, it is worth noting that the generalization bound is versatile and can be applied to a wide range of data augmentation techniques, extending beyond the scope of COCOA.

## 4.1 ASYMPTOTIC ANALYSIS

We first define a notion of consistency for data augmentation. Next, we demonstrate that the proposed consistency is equivalent to emulating RCTs. Finally, we provide convergence guarantees under the positivity assumption. Please note that our proposed analysis does not provide theoretical guarantees for the use of contrastive learning. Rather, it serves as an initial step towards understanding the effectiveness of data augmentation in CATE estimation.

**Definition 5** (Consistency of Factual Distribution). *A factual distribution $p_F$ is consistent if for every hypothesis $h : \mathcal{X} \times \{0,1\} \to \mathcal{Y}, \mathcal{L}_F(h) = \mathcal{L}_{CF}(h)$.*

**Definition 6** (Consistency of Data Augmentation). *A data augmentation method is said to be consistent if the augmented data follows a factual distribution that is consistent.*

**Proposition 1** (Consistency is Equivalent to RCT). *The factual distribution of any randomized controlled trial is consistent. More importantly, suppose we have a consistent factual distribution, then the data must originate from a distribution that is equivalent to one generating a randomized controlled trial.*

Theorem 1 suggests that any consistent data augmentation is equivalent to collecting data from an RCT - the gold standard for CATE estimation. Next, we establish the asymptotic consistency of COCOA. To this end, we demonstrate that for any given individual $x$, the likelihood of encountering neighboring data points is sufficiently high as the number of data points grows, which guarantees reliable imputation of its counterfactual outcome. This concept is formally captured in the following Theorem 2.

**Proposition 2.** *Let $x \in \mathcal{X}$ and let $\{X_k\}_{k=1}^M$ be iid samples of $X$. Under positivity, we have that for $t \in \{0,1\}$:*

$$P(\cap_{k=1}^M X_k \notin B_\epsilon(x)|T = t) \leq (1 - P(X \in B_\epsilon(x)|T = t))^M \xrightarrow[M \to \infty]{} 0$$

*where $B_\epsilon(x) = \{x' \in \mathcal{X} | \|x - x'\| < \epsilon\}$ is the $\epsilon$-ball around $x$.*

This implies that with a sufficient number of samples, the probability of not encountering data points in close proximity to any given point $x$ becomes very small. Hence, positivity ensures that within the big data regime, we will encounter densely populated regions, enabling us to approximate counterfactual distributions locally. This facilitates the application of our methods.

### 4.2 GENERALIZATION BOUND FOR DATA AUGMENTATION METHODS

Let $p_{\mathrm{F}}(x, t, y)$ denote the factual distribution of the datasets and $p_{\mathrm{CF}}(x, t, y)$ the counterfactual distribution. Counterfactual data augmentation is essentially sampling from an estimated counterfactual distribution $\widehat{p}_{\mathrm{CF}}(x, t, y)$. The distribution of the augmented factual dataset can be defined as follows:

$$p_{\mathrm{AF}}(x, t, y) = (1 - \alpha) \cdot p_{\mathrm{F}}(x, t, y) + \alpha \cdot \widehat{p}_{\mathrm{CF}}(x, t, y), \tag{10}$$

where $\alpha \in [0, \frac{1}{2}]$ represents the ratio of the number of generated counterfactual samples to the total number of samples in the augmented dataset. Let $p_{\mathrm{RCT}}(x, t, y)$ represent the distribution of $(X, T, Y)$ when the observations are sampled from randomized controlled trials.

Next, we present generalization bounds for the performance of a hypothesis trained with the augmented dataset. To establish the generalization bound, we assume that there is a true potential outcome function $f$ such that $Y = f(X, T) + \eta$ with $\eta$ verifying that $\mathbb{E}[\eta] = 0$. Let $\mathcal{A}$ denote the process of data augmentation such that $\mathcal{A}(x, t, D)$ denotes the imputed outcome for the individual $x$ under treatment $t$ where $D$ is the original dataset. Let $n = |D|$ denote the total number of samples in the original dataset. Let $\tilde{f}_n(x, t) = \mathbb{E}_D[\mathcal{A}(X, T, D)|X = x, T = t]$ denote the expected imputation when the dataset $D$ consists of $n$ samples independently sampled from $p_F$.

**Proposition 3** (Generalization Bound). *Let $\mathcal{H} = \{h : \mathcal{X} \times \{0, 1\} \rightarrow \mathcal{Y}\}$ denote all the measurable functions for potential outcome estimation. Let $\mathcal{L}_{AF} = \mathcal{L}_{p_{AF}}$ be defined as in Definition 4. Then $\tilde{f} \in \arg\min_{h \in \mathcal{H}} \mathcal{L}_{AF}(h)$. Moreover, for any measurable hypothesis function $h \in \mathcal{H}$, its $\varepsilon_{PEHE}$ is upper bounded as follows:*

$$\varepsilon_{PEHE}(h) \leq 4 \cdot \left( \mathcal{L}_{AF}(h) + V\left(p_{RCT}(X, T), p_{AF}(X, T)\right) + \alpha \cdot b_{\mathcal{A}}(n) \right) \tag{11}$$

*where $V(p, q) = \int_{\mathcal{S}} |p(s) - q(s)| ds$ is the $L_1$ distance [2] between two distributions,*

$$b_{\mathcal{A}}(n) = \mathbb{E}_{X, T \sim \widehat{p}_{\mathrm{CF}}(x, t)} \left[ \|f(X, T) - \tilde{f}_n(X, T)\|^2 \right]$$

**Interpretation of the Generalization Bound.** We first note that the first term in Theorem 3 $\mathcal{L}_{\mathrm{AF}}(h)$ is essentially the training loss of a hypothesis $h$ on the augmented dataset. The second term, which is independent of $Y$, characterizes the statistical similarity between the individuals and treatment assignments in the augmented dataset and those generated from an RCT. As there is no statistical disparity across treatment groups when $(X, T)$ follows $p_{\mathrm{RCT}}$, the closer $p_{\mathrm{AF}}$ is to $p_{\mathrm{RCT}}$ the less is the statistical disparity in the augmented dataset. Meanwhile, the third term characterizes the accuracy of the data augmentation method. Hence, this theorem provides a rigorous illustration of the trade-off between the statistical disparity across treatment groups and the imputation error. It underscores that by simultaneously minimizing disparity and imputation error, we can enhance the performance of CATE estimation models. Also note that as $\alpha$, i.e., the ratio of imputed data points to all the data points, increases, the third term increases while the second decreases. This rigorously captures another important trade-off between the precision of data imputation and the discrepancy across treatment groups. It is also essential to highlight that if the local regression module can achieve more accurate estimation with more samples (e.g., local Gaussian Process and local linear regression) $b_{\mathcal{A}}(n)$ will converge to 0 as $n$ increases. Hence, $\varepsilon_{\mathrm{PEHE}}$ will converge to 0. This result aligns with our asymptotic analysis, indicating that as the augmented dataset grows in size, it converges toward RCT. In our experiments (Section 5), we demonstrate that even small datasets can substantially benefit from augmenting the training data with just a few additional data samples using our approach.

## 5 EXPERIMENTAL STUDIES

We test our proposed methods on various benchmark datasets: the IHDP dataset (Ramey et al., 1992; Hill, 2011), the News dataset (Johansson et al., 2016; Newman et al., 2008), and the Twins dataset (Louizos et al., 2017). Additionally, we apply our methods to two synthetic datasets: one with linear functions for potential outcomes and the other with non-linear functions. A detailed description of these datasets is provided in Appendix A. To estimate the variance of our method, we randomly divide each of these datasets into a train (70%) dataset and a test (30%) dataset with varying seeds, and

---

[2] Also known as the total variation distance.

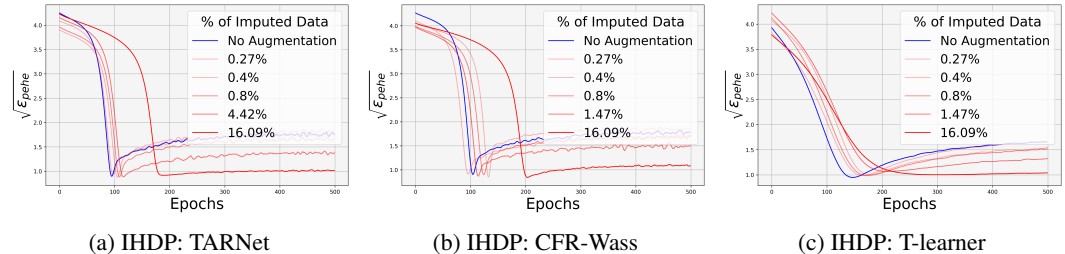

(a) IHDP: TARNet       (b) IHDP: CFR-Wass       (c) IHDP: T-learner

Figure 1: Effects of COCOA on preventing overfitting. The X-axis denotes the number of epochs, and the Y-axis represents the performance measure (not accessible in practice). As can be observed, the performance of the models trained with the original dataset without data augmentation demonstrates decreases as the epoch number increases beyond the optimal stopping epoch (blue curves), overfitting to the factual distribution. In contrast, the error of the models trained with the augmented dataset does not increase significantly (red curves), demonstrating the effect of COCOA on preventing overfitting.

Table 1: $\sqrt{\varepsilon_{\mathrm{PEHE}}}$ across various CATE estimation models, with COCOA augmentation (w/ aug.) and without COCOA augmentation (w/o aug.) on Twins, Linear, and Non-Linear datasets. Lower $\sqrt{\varepsilon_{\mathrm{PEHE}}}$ corresponds to the better performance.

| Model | Twins | | Linear | | Non-linear | |
|---|---|---|---|---|---|---|
| | w/o aug. | w/ aug. | w/o aug. | w/ aug. | w/o aug. | w/ aug. |
| TARNet | $0.59_{\pm 0.29}$ | $0.57_{\pm 0.32}$ | $0.93_{\pm 0.09}$ | $0.81_{\pm 0.02}$ | $7.41_{\pm 0.23}$ | $6.64_{\pm 0.11}$ |
| CFR-Wass | $0.50_{\pm 0.13}$ | $0.14_{\pm 0.10}$ | $0.87_{\pm 0.05}$ | $0.74_{\pm 0.05}$ | $7.32_{\pm 0.21}$ | $6.22_{\pm 0.07}$ |
| CFR-MMD | $0.19_{\pm 0.09}$ | $0.18_{\pm 0.12}$ | $0.91_{\pm 0.04}$ | $0.78_{\pm 0.06}$ | $7.35_{\pm 0.19}$ | $6.28_{\pm 0.10}$ |
| T-Learner | $0.11_{\pm 0.03}$ | $0.10_{\pm 0.03}$ | $0.90_{\pm 0.01}$ | $0.89_{\pm 0.01}$ | $7.68_{\pm 0.12}$ | $7.51_{\pm 0.07}$ |
| S-Learner | $0.90_{\pm 0.02}$ | $0.81_{\pm 0.06}$ | $0.64_{\pm 0.01}$ | $0.63_{\pm 0.01}$ | $7.22_{\pm 0.01}$ | $6.92_{\pm 0.01}$ |
| BART | $0.57_{\pm 0.08}$ | $0.56_{\pm 0.08}$ | $0.65_{\pm 0.00}$ | $0.30_{\pm 0.00}$ | $5.49_{\pm 0.00}$ | $4.50_{\pm 0.00}$ |
| CF | $0.57_{\pm 0.08}$ | $0.51_{\pm 0.11}$ | $0.63_{\pm 0.00}$ | $0.27_{\pm 0.00}$ | $5.46_{\pm 0.00}$ | $4.46_{\pm 0.00}$ |

record the results from experiments with these different splits of data. Moreover, we demonstrate the efficacy of our methods across a variety of CATE estimation models, including TARNet, CFR-Wass, CFR-MMD, T-Learner, S-Learner, BART, and Causal Forests (CF).

**Performance Improvements.** The experimental results to verify the effect of COCOA on improving the performance of CATE estimation models are summarized in Table 1 and Table 2. These tables showcase the performance of various CATE estimation models, including the state-of-the-art ones, with and without data augmentation. It can be observed that COCOA leads to consistent performance improvement for a variety of CATE estimation models. Moreover, the improvements achieved by our method are consistent across all the datasets. We also observe that COCOA significantly outperforms Perfect Match (Schwab et al., 2018) as shown in Table 2.

**Robustness Improvements.** In the context of CATE estimation, it is essential to notice the absence of a validation dataset due to the unavailability of the counterfactual outcomes. This poses a challenge in preventing the models from overfitting to the factual distribution. Our proposed data augmentation technique effectively addresses this challenge, as illustrated in Figure 1, resulting in a significant enhancement of the overall effectiveness of various CATE estimation models. Notably, counterfactual balancing frameworks (Johansson et al., 2016; Shalit et al., 2017) significantly benefit from COCOA. This improvement can be attributed to the fact that data augmentation in dense regions helps narrow the discrepancy between the distributions of the control and the treatment groups. By reducing this disparity, our approach enables better generalization and minimizes the balancing distance, leading to more stable outcomes. We include more results in Appendix C.4.

**Ablation Studies.** Additionally, we conducted ablation studies to assess the impact of two key parameters on the performance of different CATE estimation models trained with the IHDP dataset. Specifically, we investigate the influence of the epsilon ball size ($R$) in the embedding space and the number of neighbors ($K$) on the models' performance, as shown in Figure 2. These experiments illustrate the trade-off between the quality of imputation and the discrepancy of the treatment groups.

Table 2: $\sqrt{\varepsilon_{\text{PEHE}}}$ across various CATE estimation models, with COCOA augmentation (w/ aug.), without COCOA augmentation (w/o aug.), and with Perfect Match augmentation on News and IHDP datasets. Lower $\sqrt{\varepsilon_{\text{PEHE}}}$ corresponds to the better performance.

| | **News** | | **IHDP** | |
| **Model** | **w/o aug.** | **w/ aug.** | **w/o aug.** | **w/ aug.** |
|---|---|---|---|---|
| TARNet | $5.34_{\pm 0.34}$ | $5.31_{\pm 0.17}$ | $0.92_{\pm 0.01}$ | $0.87_{\pm 0.01}$ |
| CFR-Wass | $3.51_{\pm 0.08}$ | $3.47_{\pm 0.09}$ | $0.85_{\pm 0.01}$ | $0.83_{\pm 0.01}$ |
| CFR-MMD | $5.05_{\pm 0.12}$ | $4.92_{\pm 0.10}$ | $0.87_{\pm 0.01}$ | $0.85_{\pm 0.01}$ |
| T-Learner | $4.79_{\pm 0.17}$ | $4.73_{\pm 0.18}$ | $2.03_{\pm 0.08}$ | $1.69_{\pm 0.03}$ |
| S-Learner | $3.83_{\pm 0.06}$ | $3.80_{\pm 0.06}$ | $1.85_{\pm 0.12}$ | $0.86_{\pm 0.01}$ |
| BART | $3.61_{\pm 0.02}$ | $3.55_{\pm 0.00}$ | $0.67_{\pm 0.00}$ | $0.67_{\pm 0.00}$ |
| CF | $3.58_{\pm 0.01}$ | $3.56_{\pm 0.01}$ | $0.72_{\pm 0.01}$ | $0.63_{\pm 0.01}$ |
| Perfect Match | $4.09_{\pm 1.12}$ | | $0.84_{\pm 0.61}$ | |

Table 3: $\sqrt{\varepsilon_{\text{PEHE}}}$ across different similarity measures: Contrastive Learning, propensity scores, and Euclidean distance, using CFR-Wass across IHDP, News, and Twins datasets.

| **Measure of Similarity** | **Euclidean Distance** | **Propensity Score** | **Contrastive Learning** |
|---|---|---|---|
| IHDP | $3.32_{\pm 1.13}$ | $3.94_{\pm 0.21}$ | $\mathbf{0.83}_{\pm 0.01}$ |
| News | $4.98_{\pm 0.10}$ | $4.82_{\pm 0.11}$ | $\mathbf{3.47}_{\pm 0.09}$ |
| Twins | $0.23_{\pm 0.10}$ | $0.48_{\pm 0.09}$ | $\mathbf{0.14}_{\pm 0.10}$ |

The detailed ablation studies are provided in Appendix C.3. Finally, we provide ablations comparing our proposed contrastive learning method to other standard similarity measures: propensity scores and Euclidean distance in the feature space, presented in Table 3. Additionally, we include in the Appendix the results for ATE estimation C.1, as well as ablations on the performance obtained using various kernels for Gaussian processes and local linear regression C.2.

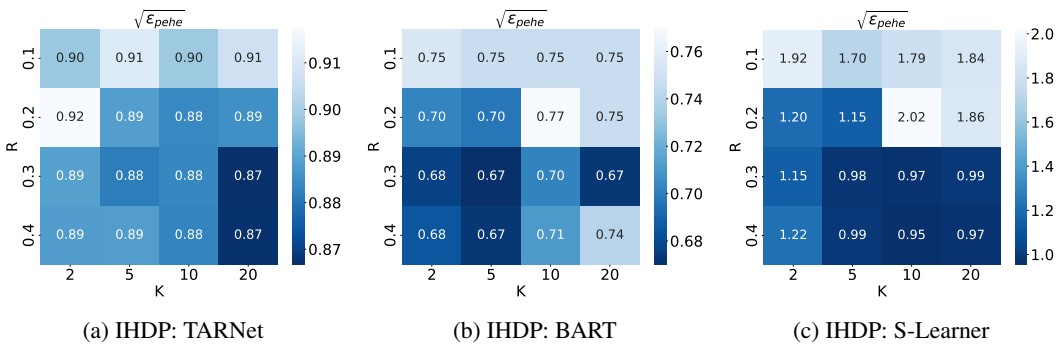

(a) IHDP: TARNet  (b) IHDP: BART  (c) IHDP: S-Learner

Figure 2: Ablation studies on the impact of the size of the $\epsilon-$Ball (R) and the number of neighbors (K) on the performance. The ablation study is conducted with three CATE estimation models on the IHDP dataset. These studies illustrate the trade-off between minimizing the discrepancy between the distributions—achieved by reducing K and increasing R—and the quality of the imputed data points, which is achieved by decreasing R and increasing K.

# 6 CONCLUSION

In this paper, we present a model-agnostic data augmentation method for CATE estimation. Our method combines theoretical guarantees with practical implementation, utilizing contrastive learning to learn a representation space where it is possible to identify a subset of individuals for which it can reliably impute their counterfactual outcomes. By adopting this approach, we enhance the performance and robustness of various CATE estimation models across benchmark datasets. We also present theoretical analysis and generalization bounds supporting our methods.

## REPRODUCIBILITY STATEMENT

The detailed dataset descriptions are provided in Appendix A, offering a comprehensive overview of the data used in our experiment. Additionally, our choice of architectures and the associated training parameters are demonstrated in Section 5.

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

APPENDIX

## A DATASET DESCRIPTIONS

**IHDP** The IHDP dataset is a semi-synthetic dataset that was introduced based on real covariates available from the Infant Health and Development Program (IHDP) to study the effect of development programs on children. The features (covariates) in this dataset come from a Randomized Control Trial. The potential outcomes were simulated following Setting B in Hill (2011). The IHDP dataset consists of 747 individuals (139 in the treatment group and 608 in the control group), each with 25 features. The potential outcomes are generated as follows:

$$Y_0 \sim \mathcal{N}(\exp(\beta^T(X + W)), 1)$$

and

$$Y_1 \sim \mathcal{N}(\beta^T(X + W) - \omega, 1)$$

where $W$ has the same dimension as $X$ with all entries equal $0.5$ and $\omega = 4$. The regression coefficient $\beta$ is a vector of length 25 where each element is randomly sampled from a categorical distribution with the support $(0, 0.1, 0.2, 0.3, 0.4)$ and the respective probability masses $\mu = (0.6, 0.1, 0.1, 0.1, 0.1)$.

**News** The News Dataset is a semi-synthetic dataset designed to assess the causal effects of various news topics on reader responses. It was first introduced in Johansson et al. (2016). The documents were sampled from news items from the NY Times corpus (downloaded from UCI Newman et al. (2008)). The covariates available for CATE estimation are the raw word counts for the 100 most probable words in each topic. The treatment $t \in \{0, 1\}$ denotes the viewing device. $t = 0$ means *with computer* and $t = 1$ means *with mobile*. A topic model is trained on a comprehensive collection of documents to generate $z(x) \in \mathbb{R}^k$ that represents the topic distribution of a given news item $x$ (Johansson et al., 2016).

Let the treatment effects be represented by $z_{c_1}$ (for $t = 1$) and $z_{c_0}$ (for $t = 0$) $z_{c_1}$ is defined as the topic distribution of a randomly selected document while $z_{c_0}$ is the average topic representation across all documents. The reader's opinion of news item $x$ on device $t$ is influenced by the similarity between $z(x)$ and $z_{c_t}$, expressed as:

$$y(x, t) = C \cdot \left( z(x)^T z_{c_0} + t \cdot z(x)^T z_{c_1} \right) + \epsilon$$

where $C = 50$ is a scaling factor and $\epsilon \sim \mathcal{N}(0, 1)$. The assignment of a news item $x$ to a device $t \in \{0, 1\}$ is biased towards the preferred device for that item, modeled using the softmax function:

$$p(t = 1|x) = \frac{e^{\kappa \cdot z(x)^T z_{c_1}}}{e^{\kappa \cdot z(x)^T z_{c_0}} + e^{\kappa \cdot z(x)^T z_{c_1}}}$$

Here, $\kappa$ determines the strength of the bias and it is assigned to be 10.

**Twins** The Twins dataset Louizos et al. (2017) is based on the collected birthday data of twins born in the United States from 1989 to 1991. It is assumed that twins share significant parts of their features. Consider the scenario where one of the twins was born heavier than the other as the treatment assignment. The outcome is whether the baby died in infancy (i.e., the outcome is mortality). Here, the twins are divided into two groups: the treatment and the control groups. The treatment group consists of heavier babies from the twins. On the other hand, the control group consists of lighter babies from the twins. The potential outcomes, $Y_0$ and $Y_1$, are generated through:

$$Y_0 \sim \mathcal{N}(\exp(\beta^T X), 0.2)$$

and

$$Y_1 \sim \mathcal{N}(\alpha^T X, 0.2)$$

Where $\beta$ and $\alpha$ are sampled from a high dimensional standard normal distribution.

**Linear dataset** We synthetically generate a dataset with $N = 1500$ samples and $d = 10$ features. The feature vectors $X = (x_1, x_2, \ldots, x_d)^T \in \mathbb{R}^d$ are drawn from a standard normal distribution. The treatment assignment $t \in \{0, 1\}$ is biased, with the probability of treatment being

$$p(t = 1 | x) = \frac{1}{1 + \exp(-(x_1 + x_2))}$$

We generate potential outcomes using two linear functions with coefficients $\beta_0 = (0.5, , \ldots, 0.5) \in \mathbb{R}^d$ and $\beta_1 = (0.3, \ldots, 0.3) \in \mathbb{R}^d$ as follows:

$$Y_0 = \beta_0 X + \mathcal{N}(0, 0.01)$$
$$Y_1 = \beta_1 X + \mathcal{N}(0, 0.01)$$

**Non-Linear dataset** We construct a synthetic dataset consisting of $N = 1500$ instances with $d = 10$ features. The feature vectors, denoted by $X = (x_1, x_2, \ldots, x_d)^T \in \mathbb{R}^d$, are sampled from a standard normal distribution. The treatment assignment $t \in \{0, 1\}$ is biased, with the probability of treatment being

$$p(t = 1 | x) = \frac{1}{1 + \exp(-(x_1 + x_2))}$$

We generate potential outcomes using two linear functions with coefficients $\beta_0 = (0.5, , \ldots, 0.5) \in \mathbb{R}^d$ and $\beta_1 = (0.3, \ldots, 0.3) \in \mathbb{R}^d$ as follows:

$$Y_0 = \exp(\beta_0 X) + \mathcal{N}(0, 0.01)$$
$$Y_1 = \exp((\beta_1 X) + \mathcal{N}(0, 0.01)$$

## B PROOFS OF THE THEORETICAL RESULTS

In this section, we include the proofs for the theoretical results presented in the main text.

**Proposition 1** (Consistency is Equivalent Randomized Controlled Trials). *Suppose we have a factual distribution $p_F$ and its corresponding counterfactual distribution $p_{CF}$ such that for every hypothesis $h : \mathcal{X} \times \{0, 1\} \to \mathcal{Y}, \mathcal{L}_F(h) = \mathcal{L}_{CF}(h)$. This implies that the data must originate from a randomized controlled trial, i.e., $p_F(X|T = 1) = p_F(X|T = 0)$.*

*Proof of Proposition 1.*
Suppose that for every hypothesis $h : \mathcal{X} \times \{0, 1\} \to \mathcal{Y}, \mathcal{L}_F(h) = \mathcal{L}_{CF}(h)$.
By definition,

$$\mathcal{L}_F(h) = \int (y - h(x, t))^2 p_F(x, t, y) \, dx \, dt \, dy$$

and

$$\mathcal{L}_{CF}(h) = \int (y - h(x, t))^2 p_{CF}(x, t, y) \, dx \, dt \, dy$$

We can write this as

$$\mathbb{E}_{p_F} \left[ (Y - h(X, T)^2) \right] = \mathbb{E}_{p_{CF}} \left[ (Y - h(X, T)^2) \right]$$

Since this holds for every function $h$, consider two Borel sets A and B in $\mathcal{X} \times \mathcal{T} \times \mathcal{Y}$, and we let $h_1(X, T) = \mathbb{E}[Y|X, T] - \mathbb{1}_A$ and $h_2(X, T) = \mathbb{E}[Y|X, T] - \mathbb{1}_B$. Hence we have that,

$$\mathbb{E}_{p_F} \left[ (Y - h_1(X, T))^2 \right] = \mathbb{E}_{p_F} \left[ (Y - \mathbb{E}[Y|X, T] + \mathbb{1}_A)^2 \right]$$

$$= \mathbb{E}_{p_F} \left[ (Y - \mathbb{E}[Y|X, T])^2 \right] + \mathbb{E}_{p_F} [\mathbb{1}_A] + 2\mathbb{E}_{p_F} [\mathbb{1}_A (Y - \mathbb{E}[Y|X, T])]$$

And we have that, $\mathbb{E}_{p_F} [\mathbb{1}_A (Y - \mathbb{E}[Y|X, T])] = 0$ since by definition of the conditional expectation we have that $\mathbb{E}[Y \mathbb{1}_A] = \mathbb{E}[\mathbb{E}[Y|X, T] \mathbb{1}_A]$. We denote by $MSE(p_F) = \mathbb{E}_{p_F} \left[ (Y - \mathbb{E}[Y|X, T])^2 \right]$. Therefore we have that

$$\mathbb{E}_{p_F} \left[ (Y - h_1(X, T))^2 \right] = MSE(p_F) + \mathbb{E}_{p_F} [\mathbb{1}_A]$$

Using the same argument for $p_{\mathrm{CF}}$ we have the following result:

$$\mathbb{E}_{p_{\mathrm{CF}}}\left[(Y - h_1(X,T))^2\right] = MSE(p_{\mathrm{CF}}) + \mathbb{E}_{p_{\mathrm{CF}}}\left[\mathbb{1}_A\right]$$

Similarly, we have the following for $h_2$:

$$\mathbb{E}_{p_{\mathrm{F}}}\left[(Y - h_2(X,T))^2\right] = MSE(p_{\mathrm{F}}) + \mathbb{E}_{p_{\mathrm{F}}}\left[\mathbb{1}_B\right]$$

$$\mathbb{E}_{p_{\mathrm{CF}}}\left[(Y - h_2(X,T))^2\right] = MSE(p_{\mathrm{CF}}) + \mathbb{E}_{p_{\mathrm{CF}}}\left[\mathbb{1}_B\right]$$

Therefore we have

$$MSE(p_{\mathrm{F}}) - MSE(p_{\mathrm{CF}}) = \mathbb{E}_{p_{\mathrm{F}}}\left[\mathbb{1}_A\right] - \mathbb{E}_{p_{\mathrm{CF}}}\left[\mathbb{1}_A\right]$$

and

$$MSE(p_{\mathrm{F}}) - MSE(p_{\mathrm{CF}}) = \mathbb{E}_{p_{\mathrm{F}}}\left[\mathbb{1}_B\right] - \mathbb{E}_{p_{\mathrm{CF}}}\left[\mathbb{1}_B\right]$$

Therefore

$$\mathbb{E}_{p_{\mathrm{F}}}\left[\mathbb{1}_A\right] - \mathbb{E}_{p_{\mathrm{CF}}}\left[\mathbb{1}_A\right] = \mathbb{E}_{p_{\mathrm{F}}}\left[\mathbb{1}_B\right] - \mathbb{E}_{p_{\mathrm{CF}}}\left[\mathbb{1}_B\right]$$

Hence it follows,

$$\mathbb{E}_{p_{\mathrm{F}}}\left[\mathbb{1}_{A \cap B}\right] = \mathbb{E}_{p_{\mathrm{CF}}}\left[\mathbb{1}_{A \cap B}\right]$$

And as this holds for every Borel measurable set $A$ and $B$, therefore we have that $p_{\mathrm{F}} = p_{\mathrm{CF}}$.

Denote by $u = p_{\mathrm{F}}(T = 1)$ we have $p_{\mathrm{F}}(X) = u p_{\mathrm{F}}(X|T = 1) + (1 - u)p_{\mathrm{F}}(X|T = 0)$. Similarly we have that $p_{\mathrm{CF}}(X) = (1 - u)p_{\mathrm{CF}}(X|T = 1) + u p_{\mathrm{CF}}(X|T = 0)$. Therefore, since $p_{\mathrm{F}} = p_{\mathrm{CF}}$,

$$u p_{\mathrm{F}}(X|T = 1) + (1 - u)p_{\mathrm{F}}(X|T = 0) = (1 - u)p_{\mathrm{CF}}(X|T = 1) + u p_{\mathrm{CF}}(X|T = 0)$$
$$= (1 - u)p_{\mathrm{F}}(X|T = 1) + u p_{\mathrm{F}}(X|T = 0)$$

Hence

$$(2u - 1)\, p_{\mathrm{F}}(X|T = 1) = (2u - 1)\, p_{\mathrm{F}}(X|T = 0)$$

Therefore we conclude the result that,

$$p_{\mathrm{F}}(X|T = 1) = p_{\mathrm{F}}(X|T = 0).$$

This concludes the proof. $\qquad\square$

For completeness, we also include this result.

**Lemma 1** (Consistency of Randomized Controlled Trials)**.** *The factual distribution of any randomized controlled trial verifying $p_F(T = 1) = p_F(T = 0)$ is consistent, i.e., if $p_F(X|T = 1) = p_F(X|T = 0)$ and $p_F(T = 1) = p_F(T = 0)$, then for all $h : \mathcal{X} \times \{0, 1\} \to \mathcal{Y}$,*

$$\mathcal{L}_F(h) = \mathcal{L}_{CF}(h)$$

*Proof.* Let $u = p_F(T = 1) = \frac{1}{2}$, $p_F(T = 1) = p_{CF}(T = 0)$

$$\mathcal{L}_{\mathrm{F}}(h) = \int (y - h(x,t))^2 p_{\mathrm{F}}(x,t,y)\, dx,\, dt\, dy$$

$$= u \int (y - h(x,1))^2 p_{\mathrm{F}}(x,y|T = 1)\, dx\, dy + (1 - u) \int (y - h(x,0))^2 p_{\mathrm{F}}(x,y|T = 0)\, dx\, dy$$

$$= u \int (y - h(x,1))^2 p_{\mathrm{F}}(x,y|T = 0)\, dx\, dy + (1 - u) \int (y - h(x,0))^2 p_{\mathrm{F}}(x,y|T = 1)\, dx\, dy$$

$$= u \int (y - h(x,1))^2 p_{\mathrm{CF}}(x,y|T = 1)\, dx\, dy + (1 - u) \int (y - h(x,0))^2 p_{\mathrm{CF}}(x,y|T = 0)\, dx\, dy$$

$$= \int (y - h(x,t))^2 p_{\mathrm{CF}}(x,t,y)\, dx\, dy$$

$$= \mathcal{L}_{\mathrm{CF}}(h)$$

$\qquad\square$

**Proposition 2** (Close neighbors are likely to appear.)**.** *Let $x \in \mathcal{X}$ and let $\{X_k\}_{k=1}^M$ be iid samples of $X$. Under positivity, i.e $\forall x' \in \mathcal{X}$, $0 < p(T = 1|X = x') < 1$, for every $\epsilon > 0$, we have that for $t \in \{0, 1\}$:*

$$P(\cap_{k=1}^M X_k \notin B_\epsilon(x)|T = t) \leq (1 - P(X \in B_\epsilon(x)|T = t))^M \xrightarrow[M \to \infty]{} 0$$

*where $B_\epsilon(x) = \{x' \in \mathcal{X}|\|x - x'\| < \epsilon\}$ is the $\epsilon$-ball around $x$.*

*Proof.* Under positivity, we have that $0 < P(X \in B_\epsilon(x)|T = t) < 1$. Hence $0 < 1 - P(X \in B_\epsilon(x)|T = t) < 1$. Therefore we conclude the result. $\square$

**Proposition 3.** *Let $\mathcal{H} = \{h : \mathcal{X} \times \{0, 1\} \to \mathcal{Y}\}$ denote all the measurable functions for potential outcome estimation. Let $\mathcal{L}_{AF} = \mathcal{L}_{p_{AF}}$ be defined as in Definition 4. Then $\tilde{f} \in \arg\min_{h \in \mathcal{H}} \mathcal{L}_{AF}(h)$. Moreover, for any measurable hypothesis function $h \in \mathcal{H}$, its $\varepsilon_{PEHE}$ is upper bounded as follows:*

$$\varepsilon_{PEHE}(h) \leq 4 \cdot \left( \mathcal{L}_{AF}(h) + V\left(p_{RCT}(X, T), p_{AF}(X, T)\right) + \alpha \cdot b_\mathcal{A}(n) \right) \tag{12}$$

*where $V(p, q) = \int_\mathcal{S} |p(s) - q(s)|ds$ is the $L_1$ distance [3] between two distributions,*

$$b_\mathcal{A}(n) = \mathbb{E}_{X,T \sim \widehat{p}_{CF}(x,t)}\left[\|f(X, T) - \tilde{f}_n(X, T)\|^2\right]$$

In order to prove Theorem 3 we start by stating a new definition for an "ideal" factual distribution. Subsequently, we will prove its consistency. The ideal factual distribution is defined as follows:

$$p_{IF} = \frac{1}{2}p_F + \frac{1}{2}p_{CF}. \tag{13}$$

In other words, to sample a dataset from $p_{IF}$, we sample from the factual distribution $p_F$ half of the time and from the counterfactual distribution $p_{CF}$ in the other half of the times. Let $p_{ICF}$ denote the counterfactual distribution corresponding to $p_{IF}$. We next show that $p_{IF}$ is consistent (thus called ideal distribution).

**Lemma 2** (Consistency of $p_{IF}$.)**.** *The error of the ideal factual distribution equals the error of its corresponding counterfactual distribution, i.e., for every hypothesis $h : \mathcal{X} \times \{0, 1\} \to \mathcal{Y}$, we have that $\mathcal{L}_{IF}(h) = \mathcal{L}_{ICF}(h)$.*

*Proof.* We observe that $p_{ICF} = \frac{1}{2}p_{CF} + \frac{1}{2}p_F$. Therefore, $p_{ICF} = p_{IF}$ and the result follows. $\square$

Intuitively, this result is saying that the ideal counterfactual augmentation gives us a factual distribution that perfectly balances the factual and counterfactual worlds. It follows from Theorem 1 that achieving this property guarantees that the dataset is identically distributed to the one generated from a Randomized Controlled Trial. However, it is impossible to sample from $p_{CF}$.

We can now prove Theorem 3.

*Proof.* We have $f : \mathcal{X} \times \{0, 1\} \to \mathcal{Y}$ to be the function underlying the true causal relationship between $(X, T)$ and $Y$ with $(X, T)$ following a distribution $p_{IF}(x, t)$. We also have $\tilde{f}_n(x, t) = \mathbb{E}_D[\mathcal{A}(X, T, D)|X = x, T = t]$ is the new function induced by the data augmentation process. Hence by construction, we have that $\tilde{f} \in \arg\min_{h \in \mathcal{H}} \mathcal{L}_{AF}(h)$. In other words, we can see $\tilde{f}_n : \mathcal{X} \times \{0, 1\} \to \mathcal{Y}$ as a new potential outcome function that generates the augmented dataset following $p_{AF}$. It follows from Theorem 3 that:

$$\mathcal{L}_{IF}(h) \leq \mathcal{L}_{AF}(h) + V(p_{IF}, p_{AF}) + \mathbb{E}_{x,t \sim p_{AF}}[\|f(x, t) - \tilde{f}(x, t)\|^2]$$

where $\mathcal{L}_{IF}$ is the factual loss with respect to the ideal density and $\mathcal{L}_{AF}$ is the factual loss with respect to the density of the augmented data.

---

[3]Also known as the total variation distance.

By decomposition of the $\varepsilon_{\text{PEHE}}$ we have that,

$$\varepsilon_{\text{PEHE}}(h) = \int_{\mathcal{X}} (h(x,1) - h(x,0) - f(x,1) + f(x,0))^2 \, p_{\text{IF}}(x) dx$$

$$= \int_{\mathcal{X}} (h(x,1) - h(x,0) - f(x,1) + f(x,0))^2 \, p_{\text{IF}}(x|T=1) p(T=1) dx dt$$

$$+ \int_{\mathcal{X}} (h(x,1) - h(x,0) - f(x,1) + f(x,0))^2 \, p_{\text{IF}}(x|T=0) p(T=0) dx dt$$

$$\leq 2 \cdot \mathcal{L}_{\text{IF}}(h) + 2 \cdot \mathcal{L}_{\text{ICF}}(h)$$

Therefore, it follows from Lemma 2 that,

$$\varepsilon_{\text{PEHE}}(h) \leq 4 \cdot \left( \mathcal{L}_{\text{AF}}(h) + V(p_{\text{RCT}}(x,t), p_{\text{AF}}(x,t)) + \mathbb{E}_{x,t \sim p_{\text{AF}}}[\|f(x,t) - \tilde{f}_n(x,t)\|^2] \right)$$

And since we have that,

$$\mathbb{E}_{x,t \sim p_{\text{AF}}}[\|f(x,t) - \tilde{f}_n(x,t)\|^2]) =$$

$$(1-\alpha) \cdot \mathbb{E}_{x,t \sim p_{\text{F}}}[\|f(x,t) - \tilde{f}_n(x,t)\|] + \cdot \alpha \mathbb{E}_{x,t \sim \widehat{p}_{\text{CF}}}[\|f(x,t) - \tilde{f}_n(x,t)\|]$$

And by observing that the first term $\mathbb{E}_{x,t \sim p_{\text{F}}}[\|f(x,t) - \tilde{f}_n(x,t)\|^2] = 0$, the result follows. ☐

**Theorem 3** (Theorem 1 in Ben-David et al. (2010)). *Let $f$ be the true function for a learning task such that $f(x) = \mathbb{E}[Y|X = x]$ where $X$ has a density $p$ and let another true function $g(x) = \mathbb{E}[Y|X = x]$ modeling another learning task, where $X$ has a density $q$. Let $h$ by a hypothesis function estimating the true function $f$, therefore we have*

$$\mathbb{E}_{X \sim q(x)}[\|g(X) - h(X)\|^2] \leq \mathbb{E}_{X \sim p(x)}[\|f(X) - h(X)\|^2] + V(p(x), p(x))$$

$$+ \mathbb{E}_{X \sim p(x)}[\|f(X) - g(X)\|^2]$$

## C  ADDITIONAL EMPIRICAL RESULTS

In this section, we present additional results for the completeness of the empirical study for COCOA. Specifically, we (*i*) study the performance of our proposed method on ATE estimation, (*ii*) conduct ablation studies on the local regression module, (*iii*) present more results for robustness against overfitting, and (*iv*) perform ablation studies on different parameters for the contrastive learning module.

### C.1  ATE ESTIMATION PERFORMANCE

In this section, we provide additional empirical results when applying our methods to ATE estimation. The error of ATE estimation is defined as:

$$\varepsilon_{ATE} = |\hat{\tau}_{ATE} - \tau_{ATE}|, \tag{14}$$

Our results are summarized in Tables 4, 5, and 6. We observe that our methods, while not tailored for ATE estimation, still bring some benefits for a subset of the estimation models.

### C.2  LOCAL REGRESSION MODULE

In this section, we compare the performance of using Gaussian Processes (GP)with different kernels vs. local linear regression. We next define the local linear regression module and present the empirical results in Table 7.

**Local Linear Regression.**  For a fixed individual $x$ who received treatment $t$, and has a selected neighbors $D_{x,t}$. Under the assumption that we can locally approximate the true function with a linear function. Suppose $X_D$ is the matrix of the observed feature values in $D_{x,t}$ augmented with a column of ones for the intercept, and $Y_D$ is the column vector of observed factual outcomes. The local linear regression coefficients, $\hat{\beta}$, are computed as:

$$\hat{\beta} = (X_D^T X_D)^{-1} X_D^T Y_D$$

Then we impute the value of $x$ as $\hat{y} = [1, x]^T \hat{\beta}$.

Table 4: $\varepsilon_{\text{ATE}}$ across various CATE estimation models, with COCOA augmentation (w/ aug.) and without COCOA augmentation (w/o aug.) in Twins, Linear, and Non-Linear datasets. Lower $\varepsilon_{\text{ATE}}$ corresponds to the better performance.

| Model | Twins | | Linear | | Non-linear | |
|---|---|---|---|---|---|---|
| | w/o aug. | w/ aug. | w/o aug. | w/ aug. | w/o aug. | w/ aug. |
| TARNet | $0.33_{\pm.19}$ | $0.41_{\pm.29}$ | $0.10_{\pm.02}$ | $0.04_{\pm.02}$ | $0.23_{\pm.13}$ | $0.04_{\pm.02}$ |
| CFR-Wass | $0.47_{\pm.16}$ | $0.14_{\pm.09}$ | $0.13_{\pm.04}$ | $0.06_{\pm.01}$ | $0.19_{\pm.09}$ | $0.03_{\pm.01}$ |
| CFR-MMD | $0.19_{\pm.09}$ | $0.18_{\pm.12}$ | $0.12_{\pm.05}$ | $0.05_{\pm.03}$ | $0.25_{\pm.15}$ | $0.04_{\pm.01}$ |
| T-Learner | $0.02_{\pm.02}$ | $0.05_{\pm.03}$ | $0.01_{\pm.01}$ | $0.01_{\pm.01}$ | $0.05_{\pm0.02}$ | $0.05_{\pm.01}$ |
| S-Learner | $0.89_{\pm.03}$ | $0.79_{\pm.07}$ | $0.03_{\pm.01}$ | $0.05_{\pm.01}$ | $0.45_{\pm.05}$ | $0.27_{\pm.02}$ |
| BART | $0.28_{\pm.08}$ | $0.21_{\pm.10}$ | $0.37_{\pm.00}$ | $0.07_{\pm.01}$ | $0.80_{\pm.00}$ | $0.26_{\pm.00}$ |
| CF | $0.28_{\pm.06}$ | $0.14_{\pm.15}$ | $0.39_{\pm.00}$ | $0.06_{\pm.01}$ | $0.77_{\pm.00}$ | $0.32_{\pm.00}$ |

Table 5: $\varepsilon_{\text{ATE}}$ across various CATE estimation models, with COCOA augmentation (w/ aug.), without COCOA augmentation (w/o aug.), and with Perfect Match augmentation in News and IHDP datasets. Lower $\varepsilon_{\text{ATE}}$ corresponds to the better performance.

| Model | News | | IHDP | |
|---|---|---|---|---|
| | w/o aug. | w/ aug. | w/o aug. | w/ aug. |
| TARNet | $0.97_{\pm.45}$ | $0.96_{\pm.38}$ | $0.12_{\pm.05}$ | $0.07_{\pm.03}$ |
| CFR-Wass | $1.00_{\pm.29}$ | $0.75_{\pm.22}$ | $0.10_{\pm.03}$ | $0.05_{\pm.02}$ |
| CFR-MMD | $0.89_{\pm.38}$ | $0.71_{\pm.22}$ | $0.16_{\pm.04}$ | $0.09_{\pm.04}$ |
| T-Learner (NN) | $0.49_{\pm.26}$ | $0.76_{\pm.20}$ | $0.27_{\pm.06}$ | $0.07_{\pm.03}$ |
| S-Learner (NN) | $0.40_{\pm.06}$ | $0.49_{\pm.27}$ | $1.72_{\pm.21}$ | $0.40_{\pm.02}$ |
| BART | $0.77_{\pm.13}$ | $0.60_{\pm.00}$ | $0.02_{\pm.01}$ | $0.02_{\pm.01}$ |
| Causal Forests | $0.72_{\pm.01}$ | $0.60_{\pm.00}$ | $0.11_{\pm.01}$ | $0.03_{\pm.02}$ |
| Perfect Match | $2.00_{\pm1.01}$ | | $0.24_{\pm.20}$ | |

## C.3 Ablation For Contrastive Learning Parameters

In this section, we provide a comprehensive set of ablation studies for the effect of the hyper-parameters of the contrastive learning module.

**Ablation on K and R.** We provide extra ablation studies on the IHDP dataset and the Non-linear dataset to study the effect of *(i)* the number of neighbors (K) and *(ii)* the embedding radius (R) on both $\varepsilon_{PEHE}$ and $\varepsilon_{ATE}$. We observe a consistently enhanced performance across different CATE estimation models. See results in figures 5 and 6. We also provide ablation studies on the sensitivity of the proposed Contrative Learning module to the parameter $\epsilon$, which is used to create the training points for the contrastive learning module by creating positive and a negative dataset, see Section 3.1 for more details.

**Ablation on the sensitivity parameter** $\epsilon$ We provide ablation on the sensitivity parameter $\epsilon$, a similarity classifier for the potential outcomes (see Section 3.1 for a detailed description). The results for the $\varepsilon_{PEHE}$ as a function of $\epsilon$ are presented in Figure 3. It can be observed that the error of CATE estimation models is consistent for a wide range of $\epsilon$, demonstrating the robustness of COCOA to the choice of hyper-parameters.

## C.4 Overfitting to the Factual Distribution

In this section, we provide more empirical results on the robustness against overfitting to the factual distribution for the Linear and Non-Linear synthetic datasets, as presented in Figure 4.

Table 6: $\varepsilon_{\text{ATE}}$ across different similarity measures: Contrastive Learning, propensity scores, and Euclidean distance, using CFR-Wass across IHDP, News, and Twins datasets.

| Measure of Similarity | Euclidean Distance | Propensity Score | Contrastive Learning |
|---|---|---|---|
| IHDP | $3.12_{\pm.33}$ | $3.85_{\pm.22}$ | $\mathbf{0.05}_{\pm.02}$ |
| News | $0.68_{\pm.20}$ | $\mathbf{0.54}_{\pm.25}$ | $0.75_{\pm.22}$ |
| Twins | $\mathbf{0.13}_{\pm.15}$ | $0.46_{\pm.09}$ | $0.14_{\pm.09}$ |

Table 7: $\varepsilon_{\text{PEHE}}$ and $\varepsilon_{\text{ATE}}$ across different local regression modules: Gaussian Process (GP) with different kernels (DotProduct, RBF, and Matern) as well as Linear Regression.

| Local Regression | GP (DotProduct) | GP (RBF) | GP (Matern) | Linear Regression |
|---|---|---|---|---|
| IHDP $\left(\sqrt{\varepsilon_{\text{PEHE}}}\right)$ | $\mathbf{0.63}_{\pm.01}$ | $0.63_{\pm.00}$ | $0.65_{\pm.02}$ | $0.75_{\pm.01}$ |
| News $\left(\sqrt{\varepsilon_{\text{PEHE}}}\right)$ | $3.56_{\pm.01}$ | $3.55_{\pm.04}$ | $\mathbf{3.44}_{\pm.05}$ | $3.53_{\pm.08}$ |
| Twins $\left(\sqrt{\varepsilon_{\text{PEHE}}}\right)$ | $\mathbf{0.51}_{\pm.11}$ | $0.51_{\pm.02}$ | $0.54_{\pm.04}$ | $0.68_{\pm.08}$ |
| IHDP $(\varepsilon_{\text{ATE}})$ | $0.02_{\pm.01}$ | $\mathbf{0.01}_{\pm.00}$ | $0.03_{\pm.01}$ | $0.09_{\pm.01}$ |
| News $(\varepsilon_{\text{ATE}})$ | $0.60_{\pm.00}$ | $0.24_{\pm.12}$ | $\mathbf{0.05}_{\pm.03}$ | $0.21_{\pm.10}$ |
| Twins $(\varepsilon_{\text{ATE}})$ | $\mathbf{0.21}_{\pm.10}$ | $0.24_{\pm.04}$ | $0.29_{\pm.04}$ | $0.38_{\pm.10}$ |

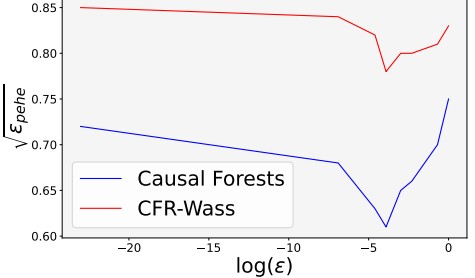 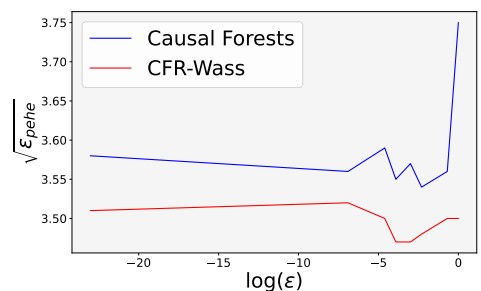

Figure 3: $\varepsilon_{\text{PEHE}}$ as a function of the similarity sensitivity parameter $\epsilon$. The figure on the left presents results for the IHDP dataset, while the one on the right is for the News dataset. Performances of two different models (CFR-Wass and Causal Forests) are plotted for both datasets.

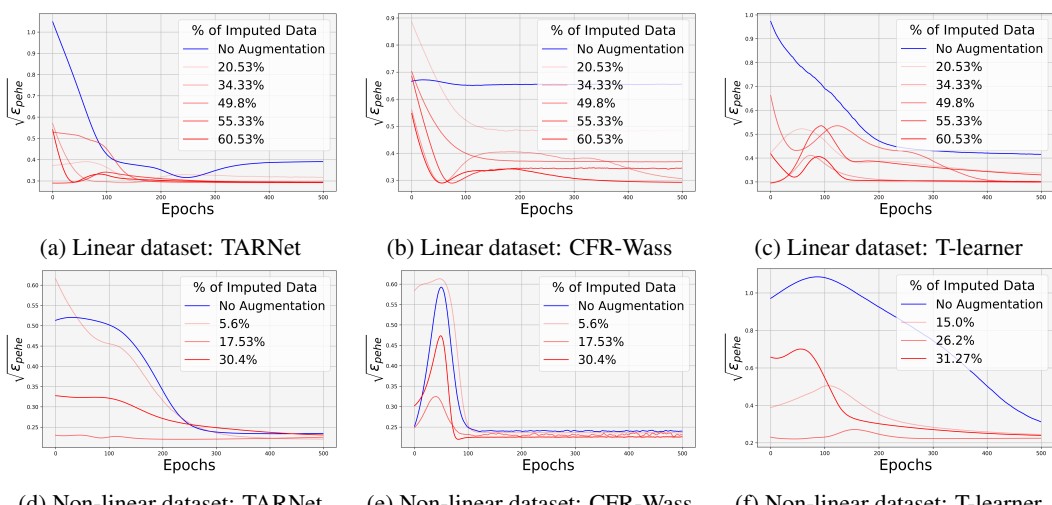

(a) Linear dataset: TARNet    (b) Linear dataset: CFR-Wass    (c) Linear dataset: T-learner

(d) Non-linear dataset: TARNet    (e) Non-linear dataset: CFR-Wass    (f) Non-linear dataset: T-learner

Figure 4: Effects of COCOA on preventing overfitting. We demonstrate the performance of three CATE estimation models trained with various levels of data augmentation on the Linear and Non-Linear datasets.

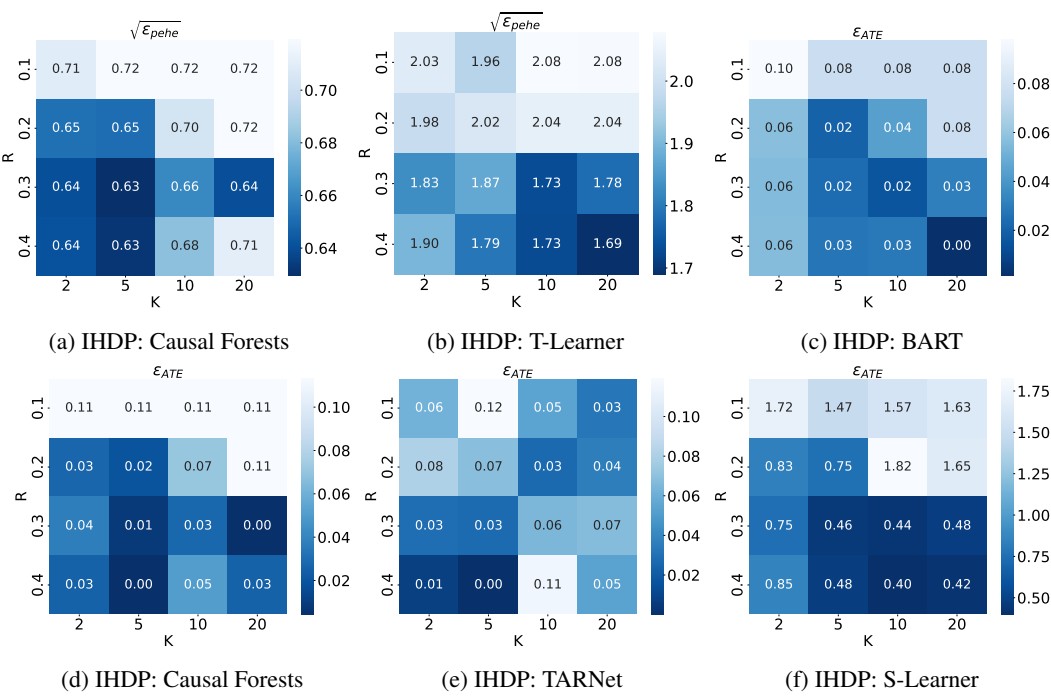

Figure 5: Ablation studies on the size of the $\epsilon-$Ball (R) and the number of neighbors (K) on the performance of different causal inference models on the IHDP dataset.

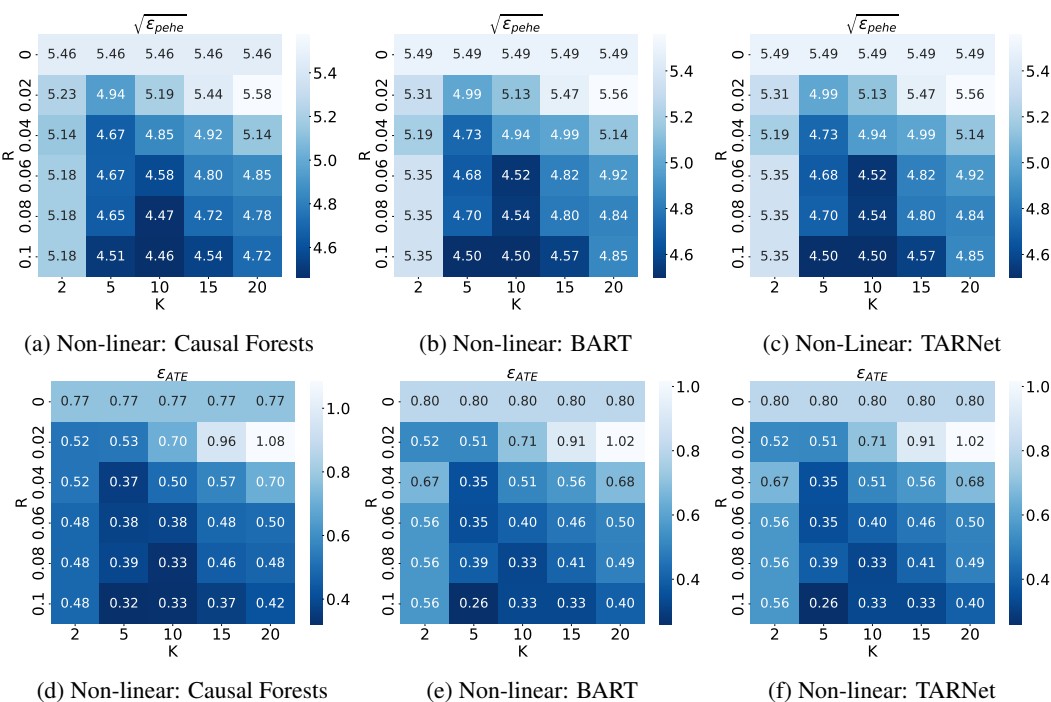

Figure 6: Ablation studies on the size of the $\epsilon-$Ball (R) and the number of neighbors (K) on the performance of different causal inference models on the Non-linear dataset.

