# OpenReview forum: "Counterfactual Data Augmentation with Contrastive Learning"
_ICLR.cc/2024/Conference — Submitted to ICLR 2024_

### Official Review · Reviewer_AQb9 · 2023-10-30

**Soundness:** 2 fair
**Presentation:** 2 fair
**Contribution:** 2 fair
**Rating:** 3
**Confidence:** 5

**Summary:**

This paper proposed a pre-processing step to Conditional Average Treatment Effects estimation. The proposed method first learn a representation space and use nearest neighbors within that space for counterfactual outcome imputation. The imputed data are used to augment the original data before piping the new data set into an existing CATE estimator.

**Strengths:**

The idea is easy to follow.

**Weaknesses:**

There are many places where the presentations are either confusing or contradictory to one another.

There are errors in the proof.

**Questions:**

1. On page 1, it stated that "consider the following experiment: for individuals in the control group (and reciprocally, the treatment group), we randomly impute their outcome under treatment (or reciprocally, in the absence of treatment), generating their counterfactual outcomes. Subsequently, we integrate each individual along with their imputed outcomes back into the dataset. This transformation ensures that individuals from both the control and treatment groups become identical, effectively eliminating any disparities". I do not agree that individuals from both groups are identical. It may be the case that the id numbers are the same, but since the missing outcomes are imputed based on some random algorithm, disparities may still exist. Disparities will only disappear when the missing outcomes are replaced by their corresponding true potential outcomes.

2. On page 3, it stated that "The counterfactual distribution, denoted by pCF, is defined as the sampling distribution of the dataset in a hypothetical parallel universe where the treatment assignment mechanism is inverted." I want to make sure that I understand it correctly that not only the treatment assignment is inverted for every individual, the observed outcome is also updated with the corresponding potential outcome under the new/inverted treatment assignment. If it's not the case, then pCF would be completed determined/induced by pF. Moreover, the proposed imputation procedure that estimates pCF also suggest that the outcome Y in pCF is different from those in pC.

    2.0 With that being said, I wonder it is true that pC(y | x,t) = pCF(y | x,t)

3. On page 4 above (5), it stated that "Let ℓ : {0, 1} × {0, 1} → R". Later, function g serves as the first arguement which has range [0,1] (the probability). Hence I wonder if it should be ℓ : [0, 1] × {0, 1} → R

4. Solving equation (5) can have two challenges. First, the number of items involve in the objective is massive, since for N subjects there are N*(N-1)/2 pairs. Second, (5) may be viewed as a binary classification problem where each pair of subjects can be viewed as an instance whose label is 1 if the pair are close or 0 if the pair are not close. From this point of view, this binary classification problem will be highly imbalanced, since most pairs are not close to each other. The authors must discuss both the computational challenge and the issue of imbalancedness.

5. Definition 5 stated that "A factual distribution pF is consistent if for every hypothesis h : X × {0, 1} → Y, LF(h) = LCF(h)".

    5.1 However, according to my understanding of pCF (see question 2 above), since pCF is not completely induced by pF, and since the computation of LCF(h) requires knowing pCF, I think this consistent property is not just a property of pF alone, but also a joint property of both pF and pCF.

    5.2 Is there any intrinsic relation between consistency and the identity mentioned earlier: pCF(x, 1 − t) = pF(x, t) (Shalit et al.,
2017; Peters et al., 2017)?

    5.3 What is the relation between consistency and unconfoundness? Does unconfoundness imply consistency or consistency imply unconfoundness?

6. Theorem 2 showed that for any given individual x, the likelihood of encountering neighboring data points is sufficiently high as the number of data points grows.

    6.1 While this theorem guarantees reliable imputation of its counterfactual outcome, it does not establish the asymptotic consistency of
COCOA (which was claimed to be the case). It only says that COCOA can estimate the counterfactual distribution well, but for consistency you need to check Definition 5.

    6.2 Theorem 2 is very trivial and the proof is very elementary. I would not call this a theorem.

7. Theorem 3, equation (3). Since the pAF is the result of a random alogorithm, depending on the data, the generalization bound in (11) can only hold with a probability, at best.

8. The definition of $b_{\mathcal{A}}(n)$ in Theorem 3 seems to be a MSE, instead of bias, to me.

9. On page 7, regarding the generalization bound, it did "underscores that by simultaneously minimizing disparity and imputation error, we can enhance the performance of CATE estimation models." But I cannot see how this provides a rigorous illustration of the trade-off
between the statistical disparity across treatment groups and the imputation error. To show a trade off you need to show how does one term increase at the cost of the decrease of another term.

10. More on the bound, I cannot see why as bA(n) will converge to 0 as n increases, εPEHE will converge to 0. There are still two more terms in εPEHE preventing it from converging to 0.

11. Table 1 in numerical studies: how to estimate εPEHE considering that the true HTE is unknown for real data sets?

12. Page 8. T-Learner and S-Learner are just general terms for groups of methods, depending on different basic learners used. Which T-Learner and S-Learner exactly did you use?

13. Page 14 in the proof, it stated that "pF(X|T = 1) = pF(X|T =0). In other words, treatment assignment is independent of the potential outcomes." But there is no potential outcome in the identity pF(X|T = 1) = pF(X|T =0). Do you mean pF(X,Y|T = 1) = pF(X,Y|T =0)?

14. Again, in Lemma 4, pF(X|T = 1) = pF(X|T = 0) does not characterize RCT. Moreover, RCT does not have to be 50-50 between treated and control, so we may not have pF(T = 1) = pF(T = 0)

15. In the proof of Lemma 4, since you only assumed that pF(X|T = 1) = pF(X|T = 0) , how did you obtain the fourth equality?

16. Again, since you only assumed that pF(X|T = 1) = pF(X|T = 0) , how did you obtain the second equality in the series of equations that started with $\mathcal{L}_{CF}(h)$?

17. At the end of the proof to Theorem 1, it stated "Therefore we have proved that for all Borel-measurable functions Φ". Unfortunately you have only proved the identity for one particular function Φ, not for all function. Therefore, the conclusion of Theorem 1 does not hold at all.

---

> ### Author Response · Authors · 2023-11-18
> **Response (Part 1)**
>
> Dear Reviewer AQb9,
>
> Thank you for your feedback. We appreciate your attention to detail and have carefully considered your comments. We updated our manuscript accordingly (see the corrected proof of theorem 1 in Appendix B). Please find below a detailed answer for the rest of the questions.
>
> ___
> > Question: “I do not agree that individuals from both groups are identical...”
> ___
>
> In the context of causal inference, when referring to the distribution of the treatment group and the control group we intend the marginals of the features $p(X|T=1)$ and p(X|T=0), not $p(Y,X|T=1)$ and $p(Y,X|T=0)$. Our characterization is accurate in illustrating that if $p(X|T=0) = p(X|T=1)$, standard learning theory guarantees that good performance on  $p(X|T=1)$ implies a good performance on prediction $Y_1$ on $p(X|T=0)$ after learning the potential outcome $Y_1$ on $p(X|T=1)$.Thus, the challenge in estimating counterfactual outcomes well lies in the disparity between the treatment groups, specifically in their feature distribution.
>
> ___
> > Question: “On page 3, it stated that "The counterfactual distribution, denoted by pCF, is defined as the sampling distribution”
> ___
>
> By inverting the treatment assignment mechanism we indeed change the structural equation relating the treatment variable T to the features X. Hence the factual outcomes will change. If the reviewer means by completely determined that there is a deterministic function between the observed factual outcome $Y = T Y1 + (1-T)Y0 $ and the counterfactual outcome random variable $Y_{CF} = (1-T) Y1 + T Y0$ then the answer is clearly yes. If he means that there is no information between both then the answer is clearly no.It is not true that $p_{F}(y|x,t) = p_{CF}(y|x,t) $ instead it is true that $p_{F}(y|x,1-t) = p_{CF}(y|x,t) $ (follows from the provided identities above).
>
> ___
> >Question: :”On page 4 above (5), it stated that "Let ℓ : {0, 1} × {0, 1} → R". Later, function g serves as the first arguement which has range [0,1] (the probability). Hence I wonder if it should be ℓ : [0, 1] × {0, 1} → R”
> ___
>
> Thank you for catching this typo, we will update the manuscript accordingly.
>
> ___
> >Question: “Solving equation (5) can have two challenges.
> ___
>
> The computational challenge and imbalance issue in solving equation (5) is indeed significant. While it is true that the number of pairs is increasing at a rate $\frac{N*(N-1)}{2}$, we only need to select a subset of all possible pairs for the training. For the imbalanced problem, we only select a subset of individuals for which we perform well, that is why we don’t augment every point in the dataset. Moreover, the regions where we have high overlap between the different treatment groups are the regions where we will do most of the imputation and these are the regions where there is the least imbalance between the different treatment groups. In practice, to overcome the imbalance problem, it is easy to sample a mini-batch with an equal number of positive and negative pairs during the learning of the contrastive learning module.
>
>
> ___
> >Question:  think this consistent property is not just a property of pF alone, but also a joint property of both pF and pCF
> ___
>
> The mentioned identity $p_{CF}(x, 1 − t) = p_{F}(x, t) $ is always true for any binary treatment assignment while consistency only holds when the treatment assignment is independent of the features $X$. There is no direct relationship between unconfoundedness, a property about observing all the features that are causally affecting $T$ and $Y$, and consistency, as defined in Definition 5, a property that is equivalent to the factual and the counterfactual distribution being identical. As an example, we can imagine a scenario where the treatment is assigned by flipping a coin without knowing any of the patient's information. We know that unconfoundedness doesn’t hold in this case (since the features are not observed) but consistency holds. Also, we can have a case where we observe all the features affecting T and Y but there is still a bias and unconfoundedness doesn’t hold. Similarly, we can construct scenarios where they both hold and both don’t.
>
> ___
> >Question:  While this theorem guarantees reliable imputation of its counterfactual outcome, it does not establish the asymptotic consistency of COCOA (which was claimed to be the case). It only says that COCOA can estimate the counterfactual distribution well, but for consistency you need to check Definition 5. 6.2 Theorem 2 is very trivial and the proof is very elementary. I would not call this a theorem.
> ___
>
> We will label it as a proposition instead. While being easy to prove, it highlights the important point that the likelihood of encountering similar individuals under positivity is increasing, which is crucial for local imputation of missing outcomes. We will clarify this in the manuscript.

---

> > ### Author Response · Authors · 2023-11-18
> > **Response (Part 2)**
> >
> > ___
> > >Question: Theorem 3, equation (3). Since the pAF is the result of a random alogorithm, depending on the data, the generalization bound in (11) can only hold with a probability, at best.
> > __
> >
> > Theorem 3's generalization bound holds for any fixed $p_{AF}$, and thus the result remains valid almost surely when considering $p_{AF}$ as a random variable.
> >
> > ___
> > >Question: The definition of $b_{\mathcal{A}(n)}$ in Theorem 3 seems to be an MSE, instead of bias, to me.
> > ___
> > We will revise the wording in Theorem 3 to clarify that it represents a squared average of bias terms.
> >
> > ___
> > >Question: On page 7, regarding the generalization bound, it did "underscores that by simultaneously minimizing disparity and imputation error, we can enhance the performance of CATE estimation models." But I cannot see how this provides a rigorous illustration of the trade-off between the statistical disparity across treatment groups and the imputation error. To show a trade off you need to show how does one term increase at the cost of the decrease of another term.
> > ___
> >
> > The local approximation assumption assumes that the potential outcomes function can locally be approximated by a linear function or a GP. Hence, for  a fixed number of data points as the radius of the ball around each individual grows the approximation becomes less accurate, this is captured by the third term in the bound in Theorem 3. On the other hand, as we increase the radius it is more likely to impute more missing potential outcomes for more individuals and thus having a closer distribution $p_{AF}(x,t)$ to a randomized controlled trials distribution $p_{RCT}(x,t)$, this is captured by the second term in the bound in Theorem 3.
> >
> > ___
> > > Question: Convergence
> > ___
> > We affirm that as all three terms in the equation converge to zero, the $e_{PEHE}$ will also converge to zero.
> >
> > ___
> > > Question: Table 1 in numerical studies: how to estimate εPEHE considering that the true HTE is unknown for real data sets?
> > ___
> > These are semi-synthetic benchmark datasets where the counterfactual outcomes are provided to test causal inference methods. Testing causal inference methods on real-world datasets is impossible as we don’t have access to the counterfactual outcomes.
> >
> > ___
> > > Question: Page 8. T-Learner and S-Learner are just general terms for groups of methods, depending on different basic learners used. Which T-Learner and S-Learner exactly did you use?
> > ___
> >
> > We specify that we use neural network architectures for both T-Learner and S-Learner. We provided a well commented code for each of the architectures.
> >
> > ___
> > > Question: Page 14 in the proof, it stated that "pF(X|T = 1) = pF(X|T =0). In other words, treatment assignment is independent of the potential outcomes." But there is no potential outcome in the identity pF(X|T = 1) = pF(X|T =0). Do you mean pF(X,Y|T = 1) = pF(X,Y|T =0)?
> > ___
> >
> > If the distributions of the control and the treatment groups are the same then this can only happen when X is independent of T. In general, it is true that T is independent of Y1 and Y0 given X but when T is also independent of X we have that T is independent of Y1 and Y0 even given the features X. We will add more details to the revised manuscript to make this clearer.
> >
> > ___
> > > Question: Again, in Lemma 4, pF(X|T = 1) = pF(X|T = 0) does not characterize RCT. Moreover, RCT does not have to be 50-50 between treated and control, so we may not have pF(T = 1) = pF(T = 0)
> > ___
> > Thank you for pointing this out, while it is true that some references use RCT to denote the general scenario when X doesn’t causally affect T, some references (e.g [1], page 92) use RCT to denote the case where the treatment is equally randomly assigned, as an ideal RCT. We will make this clearer in our updated manuscript.
> >
> > ___
> > > Question: In the proof of Lemma 4, since you only assumed that pF(X|T = 1) = pF(X|T = 0) , how did you obtain the fourth equality?
> > ___
> > It follows from the fact that $p_{CF}(X|T=1) = p_{CF}(X|T=0)$ and $p_{CF}(X|T=0) = p_{CF}(X|T=1)$ as we mentioned above. We will add this for clarity.
> >
> > In conclusion, we believe our method, primarily empirical, offers significant benefits for CATE estimation methods. We aim to build a foundation for future theoretical work to understand the limitations and potential extensions of our approach.
> >
> > Thank you once again for your valuable feedback. We hope our responses adequately address your concerns.
> >
> >
> > __________
> > References:
> >
> > [1] Friedman, Lawrence M., et al. Fundamentals of clinical trials. springer, 2015.

---

### Official Review · Reviewer_NLwi · 2023-11-01

**Soundness:** 2 fair
**Presentation:** 2 fair
**Contribution:** 2 fair
**Rating:** 5
**Confidence:** 3

**Summary:**

The work proposes a heuristic data augmentation procedure for the estimation of Conditional Average Treatment Effects. The procedure first identifies a set of candidate samples and then generates augmented data by performing local approximation on the samples.  Due to insufficient theoretical understanding, it remains unclear why the method should work.

**Strengths:**

Using contrastive learning for data augmentation for CATE estimation is novel and interesting.

**Weaknesses:**

1. There are no direct theoretical results regarding the learned representation $g_{\theta}$, the augmented data $D_{AF}$, or the local approximations. The claim that the convergence guarantees for COCOA are provided is misleading. Theorem 2 is almost trivial under the positivity assumption. Theorem 3 is a simple corollary of Theorem 7 in Appendix. They provided very limited insights into how COCOA works.

2. Ideally, there should be a principle way to decide $\varepsilon$ for generating the positive and negative datasets and $K$ for the selection of the nearest neighbors. Otherwise, the method should be extensively evaluated. More synthetic data experiments with larger classes of data distributions will help to understand when the method works or fails.

3. It could be challenging to provide finite-sample guarantees for the procedure. However, there should be experiments that examine how sensitive the method is with respect to small sample sizes.

**Questions:**

The main weakness of the procedure is the lack of theoretical guarantees that truly reveal why and when the data augmentation helps. I think the idea is promising, but additional exploration of the theoretical understanding is needed.

---

> ### Author Response · Authors · 2023-11-18
> **Response**
>
> Dear Reviewer NLwi,
>
> Thank you for your feedback.
>
> 1. Theoretical Results Significance: We respectfully disagree with the reviewer, the provided theoretical analysis establishes a foundation for understanding data augmentation methods in causal inference. Mainly Theorem 3, postulates that the error in estimating treatment effects is bounded by the loss on the training augmented data, the distance between the **features** distribution in the augmented data and that of a Randomized Controlled Trial, and the average error for the local approximator. It is important to note that the first term is guaranteed to converge to $0$ by standard learning theory, the second term is guaranteed to converge to 0 as soon as we select all the individuals to impute their outcomes, and the third term depends on the quality of imputation but is also guaranteed to converge to 0 as the number of data points increase. This being said we agree with the reviewer that further investigation of using contrastive learning to identify smooth local regions for local approximation is an open research question that requires a more theoretical understanding of contrastive learning in general. We will be happy if the reviewers can point us to good theoretical works for contrastive learning that will help us in this research direction. We will also change the wording in our updated manuscript to make the limitations of our preliminary theoretical analysis clearer.
>
> 2. Hyperparameters $\epsilon$ and $K$: We agree with the reviewer that these hyperparameters are very important for the efficacy of the method, which is why we conducted **extensive ablation studies** on them presented in the main text and in Appendix C.3. We conducted our experiments in benchmark causal inference datasets (IHDP, Twins, and News) as well two synthetic datasets (a Linear and Nonlinear dataset). Our experimental results suggest that our method is robust to a wide range of choices of hyper-parameters: it is not challenging to find a set of hyper-parameters that consistently improve the performance. This being said, we believe developing principal ways to tune these hyperparameters (as well as other widely used hyperparameters in Machine Learning in general and causal inference in particular) is a very important research direction that is beyond the scope of the current presentation.
>
> 3. Sample Size Effect: Semi-synthetic datasets such as IHDP already have a limited number of individuals, and reducing the dataset size further complicates the learning process, particularly for data augmentation methods. As the dataset shrinks, finding close neighbors becomes increasingly challenging. Consequently, when the number of data points is very small, the method may struggle to identify suitable neighbors, potentially leading to no data points being selected for data augmentation. However, we would like to note that compared to other methods, our algorithm will not perform any data augmentation in this case. This is one of the key properties of our algorithm that it only imputes with high confidence, thus never worsening the problem.
>
> We thank the reviewer again for his feedback and we will be happy to answer any further questions.

---

> > ### Comment · Reviewer_NLwi · 2023-11-19
> >
> > Let me clarify the concern regarding the lack of theoretical guarantees in this work. Intuitively, generating more samples at places where the samples are dense does not necessarily make the empirical distribution of the samples closer to the true distribution. There are many settings where this data augmentation strategy may fail. For instance, when the dimension of $X$ is relatively high, there could many sparse regions. Only generating samples for dense regions will lead to poor estimations. The authors should provide theoretical justification for this data augmentation strategy. This will help the readers see when the proposed method is applicable. Otherwise, if there is a rich literature on this strategy, please use it to support your work.
> >
> > The contrastive learning method is simple, I do not think theoretical guarantees have to be provided.
> >
> > I am pleased with the response to my other questions. Based on the response, I adjust my score, but I keep my decision.

---

> > > ### Author Response · Authors · 2023-11-21
> > > **Response**
> > >
> > > Dear Reviewer NLwi,
> > >
> > > Our algorithm is designed to analyze situations where there's a group that received treatment (with a distribution denoted as $ p_F(X|T=1)$) and a group that did not (with a distribution  $p_F(X|T=0)$). To estimate the outcomes for the treatment group as if they were in the control group, we select a subset from the treatment group and impute their control outcomes. As we do this, the distribution of the treatment group gradually transforms to $p_{AF}(X|T=1) = \alpha p_F(X|T=1) + (1-\alpha) p_F(X|T=0) $. Essentially, the more we impute, the closer we get to mimicking a distribution one would expect in a randomized controlled trial. That said, we agree with the reviewer that further investigation into the finite sample scenario, where only a small subset of individuals in dense regions have been imputed, and developing theoretical guarantees for it is an important research direction. We intend to support our empirically successful proposed algorithm with a more detailed theoretical understanding of local approximation in causal inference in our future work.
> > >
> > > Thank you again for your valuable feedback.

---

### Official Review · Reviewer_Lde6 · 2023-11-02

**Soundness:** 3 good
**Presentation:** 3 good
**Contribution:** 3 good
**Rating:** 5
**Confidence:** 5

**Summary:**

A data augmentation technique to improve accuracy of causal effect estimation. A method is proposed to augment units with their estimated potential outcomes. Such an augmentation can help any downstream effect estimator by increasing overlap.

**Strengths:**

- Good idea on using data augmentation based on a learnt representation.
- Can apply to all dowstream methods
- An attempt to characterize theoretical bounds of the method

**Weaknesses:**

- The main technique is to apply contrastive learning. However, the loss does not align with the causal effect problem. Ideally, we would like the individuals with the same causal effect to be close to each other. But the proposed method constraints individuals with the same outcome (under same treatment) to be close to each other. I understand that the former is not identified from observed data, but stating this distinction is important. As a result, I do not see a conceptual justification on why the proposed method will work well. It is easy to generate a counter-example. E.g., consider a dataset where treatment only affects young people, but not old people. Assume outcome values are as follows:
Young: (T=0, Y=0), (T=1, Y=10)
Old: (T=0, Y=0), (T=1, Y=0)
Now the proposed algorithm will move untreated young and old people together. For T=1, it will also have young and old people as negative examples. The net result will be that the feature "age" in the representation will be significantly weakened.

-  The theory is obfuscating the real issue. While the theory provides bounds based on quality of augmented data, the key concern is the quality of data generated by proposed algorithm (and whether it is likely to be close to the randomized distribution). So the theory does not provide any more confidence in the method's soundness--it could have been written for any data augmentation method.

- I would suggest that the authors compare to more state-of-the-art methods, such as ReiszNet. https://arxiv.org/abs/2110.03031 Would the stated gains be still significant?

Finally, it may be useful to connect to existing literature on data augmentation and matching in causal ML. Some references:
1. https://icml.cc/virtual/2021/spotlight/8888 selecting data augmentation for simulating intervention
2. http://proceedings.mlr.press/v139/mahajan21b.html

**Questions:**

- Justify the contrastive loss.
- Can you compare to ReiszNet?
- The selection criterion uses g(xi,xj)=1. Can the similarity be exactly one, or do you use a threshold in practice?

---

> ### Author Response · Authors · 2023-11-18
> **Response**
>
> Dear Reviewer Lde6,
>
> Thank you for your constructive feedback and for the relevant resources you shared with us. We will include them in our revised related work section.
>
> 1. The contrastive loss: Contrastive Learning is used as a heuristic (proven to be effective in various applications) to identify similar individuals **within the same treatment group**. To clarify the potential misunderstanding, as in the provided example the pair $\{X = \text{“young”}, T=0, Y=0\}$ and $\{X = \text{“old”}, T=0, Y=0\}$ will be in the positive dataset. And the pair $\{X = \text{“young”}, T=1, Y=10\}$ and $\{X = \text{“old”}, T=1, Y=0\}$ will be in the negative dataset. In this case, no data augmentation will happen. This is indeed the key to the success of our algorithm: it does not impute unreliable outcomes. Assuming that we have more points. Therefore if we get individual $a=\{X = \text{“old”}, T=1, Y=0\}$ (Y value is hidden from us), we can test that it will be similar to the individuals $b=\{X = \text{“old”}, T=0, Y=0\}$ and we can hence impute his outcome and have $Y \approx 0$ (a has the same $X$ value as $b$ hence $b$ is a close neighbor of $a$, and we use $b$’s factual outcome as the imputation results for $a$).
>
> 2. The place of the theory: The proposed theoretical analysis applies to general data augmentation in causal inference. While local approximators are guaranteed to converge (there has been a considerate amount of work that studies the statistical consistency of local approximation methods as in [1], hence we believe it is unnecessary to repeat these results in this work), the contrastive learning part is a heuristic that we use to identify similar individuals. The other natural choice is using Euclidean distance but that doesn’t generalize well for high dimensional data. We will add a remark to our revised manuscript outlining this.
>
> 3. Comparison to ReiszNet: To the best of our understanding the provided reference builds a method to estimate the average treatment effect (ATE). Our proposed method is mainly built to estimate the conditional average treatment effect (CATE) at an individual level. Hence, we fail to see a direct way of applying Reisznet to CATE estimation. We would appreciate it if the reviewer could share advice on how to adapt ReiszNet to CATE estimation. This being said, we will include this method for completeness in the related work when discussing ATE estimation methods.
>
> 4. Selection Criteria: Yes we use the same threshold $\epsilon$ in practice, hence if $\|g(x_i,x_j)\|\leq \epsilon$ we classify the individuals as similar otherwise we classify them as non-similar. We will add this detail to the algorithm section for enhanced clarity.
>
> Thank you again for your constructive feedback.
>
> [1] Tjøstheim, D., Otneim, H. and Støve, B., 2021. Statistical Modeling Using Local Gaussian Approximation. Academic Press.

---

### Official Review · Reviewer_FUqM · 2023-11-05

**Soundness:** 3 good
**Presentation:** 3 good
**Contribution:** 3 good
**Rating:** 6
**Confidence:** 3

**Summary:**

This paper proposes a framework named COCOA which aims to estimate the conditional average treatment effects (CATE) by imputing the counterfactual outcomes for a subset of the subjects. The subset to be imputed is selected by learning a representation space and a similarity measure with the assumption that in the learned representation space close individuals identified by the learned similarity measure have similar potential outcomes. Theoretical analysis is performed and empirical experiments show improved performance.

**Strengths:**

- The paper is overall well-written and the development of the main ideas is easy to follow.
- The idea of performing counterfactual outcome imputation over a subset of individuals guided by a learned latent space is quite interesting and seems novel.
- Although the contrastive learning algorithm used in this paper is not new, its use in the overall learning framework is reasonable.
- The proposed method is model-agnostic, making it possible to be plugged in for any CATE estimation model, including the potentially better ones to be developed in the future.
- The paper theoretically builds the generalization bound and provides insights into the theoretical development.
- Empirical evaluation shows significant improvement by combining the proposed augmentation method with the existing CATE estimation methods.

**Weaknesses:**

- No analysis of the computational complexity is provided. It is known how the proposed method scales with large datasets.
- The pairs of similar individuals are defined as individuals having a difference of outcomes less or equal to a threshold. When applied to data with binary outcomes (e.g., product purchase or not), the positive dataset $D^+$ will potentially contain too many individuals. It is unclear if this will have a negative impact on the proposed method and how this can be further addressed.
- Definition 4 defines a factual loss and a counterfactual loss, which takes the integral over time, implying that a continuous-time loss/model is used. However, the exact form of the loss function used in later sections (except the theoretical analysis) seem to imply the opposite. It would be great to clarify how and why Definition 4 is used.

**Questions:**

- What is the computational complexity of the proposed method?
- Can the method apply to datasets with binary outcomes? How to reduce the size of $D^+$ in this case?
- How and why is Definition 4 used in the learning procedures?

---

> ### Author Response · Authors · 2023-11-18
> **Response**
>
> Dear Reviewer FUqm,
>
> Thank you for your valuable feedback.
>
> 1. Computational Complexity: The computational complexity of identifying the similar and the non-similar individuals is $\frac{N^2}{2}$ in the worst-case scenario. The complexity of the rest of the learning algorithm is on par with that of the other modules. For very large datasets, we can select a subset of the possible similar and non-similar pairs. In practice, when the pairs of positive samples are moderate, we can select only a small set of negative samples for every gradient update of the contrastive learning module. Please note that when the dataset contains lots of pairs of similar individuals, data augmentation is not needed as performance is guaranteed by our error bound (simply setting $\alpha = 0$).
>
> 2. Binary Outcomes: Since we assume that the functions are locally smooth and can be approximated locally by a linear model or a Gaussian Process, the method currently does not generalize well to classification tasks. We will incorporate this clarification in our revised manuscript.
>
> 3. Definition 4:  Definition 4 establishes the counterfactual and factual losses, which are crucial in the proposed theoretical analysis in Section 4. Please note that $t$ in the definition is the treatment assignment (that is discrete in our case). This analysis introduces a notion of consistency for data augmentation in causal inference.
>
> We hope these clarifications address your concerns and reinforce the value of our work. Thank you once again for your insightful comments.

---

### Meta-Review · Area_Chair_21fu · 2023-12-09

**Metareview:**

The presented work introduces a comprehensive approach to estimating Conditional Average Treatment Effects (CATE). It proposes a heuristic data augmentation procedure, identifying a set of candidate samples and generating augmented data through local approximation. Despite a lack of thorough theoretical understanding regarding the method's efficacy, it stands as an interesting proposal. The pre-processing step for CATE estimation involves learning a representation space and employing nearest neighbors for counterfactual outcome imputation. The imputed data augment the original dataset before being input into an existing CATE estimator. Empirical experiments showcase improved performance.


The authors are encouraged to strengthen their theoretical analyses. Specifically, the reviewers pointed that the presented theoretical results shadowed the requirement of the quality of the augmented data. There are also a number of identified inconsistencies in the presented analysis that the authors can use another round of revision to clean up. There is also a main concern about the use of contrastive loss, which does not fully align with the goal of the problem.

**Justification For Why Not Higher Score:**

The paper has potential but as of now, it has a number of issues the authors can better clarify. The authors are also encouraged to better justify their design choice and double check their analytical results.

**Justification For Why Not Lower Score:**

N/A

---

### Decision · Program_Chairs · 2024-01-16

Reject